# Probabilistic Inference with Algebraic Constraints: Theoretical Limits and Practical Approximations

**Zhe Zeng**[*]
CS Department
UCLA
zhezeng@cs.ucla.edu

**Paolo Morettin**[†]
DTAI
KU Leuven, Belgium
paolo.morettin@unitn.it

**Fanqi Yan**[*]
CS Department
UT Austin
fqyan@cs.utexas.edu

**Antonio Vergari**
CS Department
UCLA
aver@cs.ucla.edu

**Guy Van den Broeck**
CS Department
UCLA
guyvdb@cs.ucla.edu

## Abstract

Weighted model integration (WMI) is a framework to perform advanced probabilistic inference in hybrid domains, i.e., on distributions over mixed continuous-discrete random variables and in the presence of complex logical and arithmetic constraints. In this work, we advance the WMI framework on both the theoretical and algorithmic side. First, we trace the boundaries of tractability for WMI inference in terms of two key properties of a WMI problem's dependency structure: sparsity and diameter. We prove that exact inference is only efficient if that structure is tree-shaped with logarithmic diameter. While this result deepens our theoretical understanding of WMI it hinders the practical applicability of exact WMI solvers to large problems. To overcome this, we propose the first approximate WMI solver that does not resort to sampling, but performs exact inference on an approximate model. Our solution iteratively performs message passing in a relaxed problem structure to recover lost dependencies. As our experiments show, it scales to problems that are out of the reach of exact WMI solvers while delivering accurate approximations.

## 1 Introduction

Consider an autonomous agent operating under uncertainty in a real-world scenario, for instance a self-driving vehicle. It has to model both *continuous* variables like the speed and position of other cars and *discrete* ones like the color of traffic lights and the number of pedestrians. Moreover, in order to make decisions, it needs to perform advanced probabilistic reasoning. For example, it has to reason about physical constraints while computing the probability of a grounded scene described via complex *algebraic constraints*, such as the geometry of vehicles and the roads ahead.

Performing probabilistic inference in these constrained and hybrid (mixed continuous-discrete) scenarios goes beyond the limited inference capabilities of intractable probabilistic models such as variational autoencoders [28] and generative adversarial networks [25]. This is also the case for classical probabilistic graphical models for hybrid domains [27, 32] and more recent tractable alternatives [33, 38, 40] which struggle to either perform inference over complex algebraic constraints or make too simplistic representational or distributional assumptions.

---

[*]Authors contributed equally. This research was performed while F.Y. and P.M. were visiting UCLA.
[†]This work was partially carried out when P.M. was working at the University of Trento.

On the other hand, Weighted Model Integration (WMI) [8, 34] is a modeling and inference framework that supports general hybrid probabilistic reasoning over algebraic constraints, *by design*. Indeed, in the WMI framework, mixed complex continuous-discrete interactions can be easily expressed in the language of Satisfiability Modulo Theories (SMT) [7] and answering probabilistic queries involving algebraic constraints can be naturally cast as integration of certain weight functions over the regions that satisfy those constraints.

In this paper we advance the WMI framework on two fronts. First, we deepen the theoretical understanding of the complexity of WMI inference on real-world problems by proving hardness results. Second, we deliver an efficient and accurate approximate WMI solver as a practical algorithmic solution to deploy WMI inference at a larger scale.

Specifically, we study the dependency structure of WMI problems as specified by the primal graph of their SMT formula [22]. We prove that performing exact inference is #P-hard if the primal graph has a treewidth larger than one or a diameter that is linear in the number of variables. Second, to overcome these negative results, we introduce RECOIN , a practical algorithmic solution that extends the relax-compensate-and-recover framework [14, 16, 17] for approximate discrete inference to hybrid inference scenarios with algebraic constraints. As our experiments suggest RECOIN candidates as the best alternative, in terms of scalability and accuracy of the delivered approximations, in the current panorama of general-purpose WMI solvers.

The rest of the paper is organized as follows. In Section 2 we introduce the notation and background needed to later prove our theoretical results in Section 3 and to introduce RECOIN in Section 4. Before evaluating RECOIN in Section 6 we discuss related work in Section 5.

## 2    Background

**Notation.**    Uppercase letters denote random variables $(X, B)$ and lowercase letters denote their assignments $(x, b)$. We use bold for sets of variables $(\mathbf{X}, \mathbf{B})$, and their joint assignments $(\boldsymbol{x}, \boldsymbol{b})$. We use capital Greek letters for logical formulas $(\Gamma, \Delta)$. Literals are atomic formulas or their negation, and are denoted using either $\ell$ or lowercase Greek letters $(\gamma, \delta)$. We let $\boldsymbol{x} \models \Delta$ denote the satisfaction of a formula $\Delta$ by an assignment $\boldsymbol{x}$. Its corresponding indicator function is $[\![\boldsymbol{x} \models \Delta]\!]$.

**Satisfiability Modulo Theories .**    To represent complex relationships between discrete and continuous variables, we harness the language of Satisfiability Modulo Theories (SMT) [7] which generalizes Boolean propositional logic [6]. Specifically, we use SMT over linear real arithmetic ($\mathcal{LRA}$) which has been used as an expressive modeling language for probabilistic programming [13], model checking [23] and robotics [20]. As is common, we adopt quantifier-free SMT($\mathcal{LRA}$) formulas and we assume them to be in conjunctive normal form (CNF), that is, a conjunction of clauses. For brevity, we will refer to them as simply SMT formulas. To characterize the dependency structure of an SMT formula we make us of its *primal graph* representation.

**Definition 2.1.**  *(Primal Graph) Let $\Delta$ be an SMT formula. Then its primal graph $\mathcal{G}_\Delta = (\mathcal{V}, \mathcal{E})$ is the undirected graph whose vertex set $\mathcal{V}$ is the set of variables in formula $\Delta$, and whose edge set $\mathcal{E}$ has edge $X - Y$ iff variable $X$ and variable $Y$ appear together in one clause $\Gamma \in \Delta$.*

**Example 2.2** (SMT formula and its primal graph). *Consider the SMT formula $\Delta$ on the left over continuous variables $X, Y, Z$ and boolean variable $B$, its primal graph $\mathcal{G}_\Delta$ is shown on the right.*

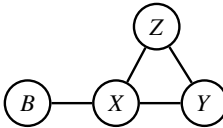

**Weighted Model Integration (WMI).**    Weighted Model Integration (WMI) [8, 34] is a framework for probabilistic modeling and inference over mixed continuous-discrete distributions in presence of algebraic constraints defined as SMT formulas. These representations are captured by WMI models.

**Definition 2.3.**  *(WMI model) Let $\mathbf{X}$ be a set of continuous random variables assuming values in $\mathbb{R}$, and $\mathbf{B}$ a set of Boolean random variables assuming values in $\mathbb{B} = \{\texttt{true}, \texttt{false}\}$. A WMI model is a pair $(\Delta, w)$, where $\Delta$ is an SMT formula over $\mathbf{X}$ and $\mathbf{B}$, and $w : (\boldsymbol{x}, \boldsymbol{b}) \mapsto \mathbb{R}^+$ is a positive function, called the weight function.*

We consider classes of WMI problems whose weight function comes from a parametric function family, denoted $\mathbf{\Omega}$. Moreover, we adopt the common assumption of weight functions $w$ to be defined as products of per-literal weights [8, 11, 41]. That is, $w$ is definable via a set of functions $\mathcal{W} = \{w_\ell(\boldsymbol{x})\}_{\ell \in \mathcal{L}}$, where $\mathcal{L}$ are the literals in $\Delta$. and where each $w_\ell$ is defined over variables in literal $\ell$. Then, the weight of assignment $(\boldsymbol{x}, \boldsymbol{b})$ is: $w(\boldsymbol{x}, \boldsymbol{b}) = \prod_{\ell \in \mathcal{L}} w_\ell(\boldsymbol{x}, \boldsymbol{b})^{[\![\boldsymbol{x}, \boldsymbol{b} \models \ell]\!]}$. Hence, we will represent WMI models as pairs $(\Delta, \mathcal{W})$.

**Definition 2.4.** *(WMI task) Let $(\Delta, \mathcal{W})$ be a WMI model over real variables $\mathbf{X}$ and Boolean variables $\mathbf{B}$. The* WMI *task for $(\Delta, \mathcal{W})$ is to compute*

$$\text{WMI}(\Delta, \mathcal{W}; \mathbf{X}, \mathbf{B}) \triangleq \sum_{\boldsymbol{b} \in \mathbb{B}^{|\mathbf{B}|}} \int_{(\boldsymbol{x}, \boldsymbol{b}) \models \Delta} \prod_{\ell \in \mathcal{L}} w_\ell(\boldsymbol{x}, \boldsymbol{b})^{[\![\boldsymbol{x}, \boldsymbol{b} \models \ell]\!]} \, d\boldsymbol{x}. \tag{1}$$

*That is, the task is to sum over all possible Boolean assignments $\boldsymbol{b} \in \mathbb{B}^{|\mathbf{B}|}$ while integrating over the weighted assignments of $\mathbf{X}$ that satisfy the formula: $(\boldsymbol{x}, \boldsymbol{b}) \models \Delta$.*

When all weights $w_\ell(\boldsymbol{x})$ are constants and all variables continuous ($\mathbf{B} = \emptyset$) we retrieve the model integration (MI) task [41], whereas when all variables are Boolean (i.e., $\mathbf{X} = \emptyset$) WMI equals the well-known weighted model counting (WMC) task [11]. In the general case, solving $\text{WMI}(\Delta, \mathcal{W}; \mathbf{X}, \mathbf{B})$ equals to computing the partition function of the unnormalized probability distribution induced by weights $\mathcal{W}$ on formula $\Delta$ and restricted to the regions where $\Delta$ is SAT.

As such, computing the probability of an event represented as an SMT formula $\Phi$ involving algebraic constraints w.r.t. the distribution induced by $\mathcal{W}$ on $\Delta$ can be done by computing the WMI of the conjunction of formula $\Delta$ and formula $\Phi$, normalized by the partition function:

$$\text{Pr}_\Delta(\Phi) = \text{WMI}(\Delta \wedge \Phi, \mathcal{W}; \mathbf{X}, \mathbf{B}) / \text{WMI}(\Delta, \mathcal{W}; \mathbf{X}, \mathbf{B}).$$

**Example 2.5** (Advanced probabilistic inference with WMI). *Consider the SMT formula $\Delta$ in Example 2.2 with per-literal weights $\mathcal{W} = \{w_{\ell_1}(B) := 2; \ w_{\ell_2}(x) := x^2; \ w_{\ell_3}(y, z) := 2yz\}$ where $\ell_1 := B, \ell_2 := x \geq 1, \ell_3 := y + z \leq 3$ and all the weights associated to other literals are constantly 1. Then the WMI of formula $\Delta$ evaluates to:*

$$\text{WMI}(\Delta, \mathcal{W}; \mathbf{X}, B) = \int_1^2 dx \int_1^{-x+3} dy \int_{-x+2}^{-y+3} x^2 \cdot (2 + 1) \cdot (x + y) \cdot 2yz \, dz = \frac{11173}{480}.$$

*Moreover, for the two formulas $\Phi_c = (B = true)$ and $\Phi_1 = 0 \leq z \leq 1$, then*

$$\text{Pr}_\Delta(\Phi_1 | \Phi_c) = \text{WMI}(\Delta \wedge \Phi_c \wedge \Phi_1, \mathcal{W}; \mathbf{X}, B) / \text{WMI}(\Delta \wedge \Phi_c, \mathcal{W}; \mathbf{X}, B) = 18936 / 78211 \approx 0.242.$$

From here on, w.l.o.g. we will assume WMI problems to be defined on continuous variables only. We leverage the polytime reduction introduced in Zeng and Van den Broeck [41] to map a WMI problem $(\Delta, \mathcal{W})$ over continuous and Boolean variables $\mathbf{X}$ and $\mathbf{B}$ to a new WMI problem $(\Delta', \mathcal{W}')$ over continuous variables $\mathbf{X}'$ only. This is done by properly introducing auxiliary variables in $\mathbf{X}'$ to account for $\mathbf{B}$. The resulting primal graph $\mathcal{G}_{\Delta'}$ is isomorphic to $\mathcal{G}_\Delta$. For instance, we can replace $B$ in Example 2.5 by a real variable $T_B$ having values in $[-1, 1]$ without changing the WMI task nor the treewidth or the diameter of the primal graph (cf. Appendix A).

## 3 On the hardness of WMI

While the general formulation of WMI we have provided in the previous section is elegant and appealing for advanced probabilistic reasoning, it is, however, not practical in general. In fact, it requires solving an arbitrarily complex integral, which is a #P-hard problem [5].

To fill this gap, recent works have started looking for *classes of tractable WMI problems*, i.e., problems for which a solution can be computed exactly in polytime [41, 42]. These classes of problems can be characterized by two parameters: the *treewidth* and the *diameter* of the primal graph of the SMT formulas considered, where the latter is generally expressed as a function of the number of variables in the problem. Note that this is strikingly different from classical discrete probabilistic graphical models, where most of the complexity results are stated in terms of the treewidth alone [31, 36].

**Definition 3.1.** *($\mathcal{WMI}(\mathbf{\Omega}, \delta, t)$ Problem Class) Let $\mathcal{WMI}(\mathbf{\Omega}, \delta, t)$ be the class of WMI problems over models of the form $(\Delta, \mathcal{W})$ on real domains, having primal graph $\mathcal{G}_\Delta$ with diameter of $\Theta(\delta(n))$ and treewidth t, where n is the number of variables in the formula $\Delta$; and having per-literal weights $\mathcal{W}$ in a function family $\mathbf{\Omega}$.*

The largest tractable WMI class known so far has been introduced in Zeng et al. [42] as $\mathbb{WMI}(\mathbf{\Omega}, \log(n), 1)$, i.e., the class of problems over $n$ real variables whose primal graph is tree-shaped (treewidth 1) and has diameter of length logarithmic in $n$, and whose weight functions belong to a function family $\mathbf{\Omega}$ satisfying some conditions called *tractable weight conditions* (TWCs).

**Definition 3.2.** *(TWCs) Given a parametric weight function family $\mathbf{\Omega}$, it satisfies the TWCs iff*

    *i) it is closed under product, i.e., $\forall f, g \in \mathbf{\Omega}, f \cdot g \in \mathbf{\Omega}$;*
    *ii) it is closed under definite integration, i.e., $\forall f \in \mathbf{\Omega}, F(u(x)) - F(l(x)) \in \mathbf{\Omega}$ where $F$ is the antiderivative of $f$, and $l(x), u(x)$ are SMT($\mathcal{LRA}$) integration bounds for any $x \in \mathbf{X}$;*
    *iii) the symbolic antiderivative of any $f \in \mathbf{\Omega}$ can be tractably computed by symbolic integration.*

Examples of weight functions in family $\mathbf{\Omega}$ include the largely adopted family of (piecewise) polynomials [8], the family of exponentiated linear functions and the family of their products. In the following analysis, we will restrict our attention to weight function families satisfying the TWCs.

In Zeng et al. [42] the tractability of problem class $\mathbb{WMI}(\mathbf{\Omega}, \log(n), 1)$ is demonstrated by construction, where they introduce a message passing scheme, named MP-WMI, that runs in polytime on tree-shaped and diameter-bounded primal graphs. That is, some *sufficient* conditions for tractable WMI classes are provided. Here we provide a finer charting of the "tractable islands" of WMI problems by questioning the necessity of the above conditions while looking for larger tractable classes. We prove that unless $P = NP$, larger classes are not tractable. We begin by proving that increasing the diameter of a tree-shaped problem structure makes it hard.

**Theorem 3.3.** *Let $\mathbb{WMI}(\mathbf{\Omega}, n, 1)$ be the class of WMI problems whose weight function family $\Omega$ satisfies the TWCs. Then inference in $\mathbb{WMI}(\mathbf{\Omega}, n, 1)$ is #P-hard.*

*Sketch of proof.* We build a polytime reduction from a #P-complete variant of the subset sum problem [24, 12, 26] to a WMI problem with constant weights and whose primal graph $\mathcal{G}_\Delta$ is a chain with diameter exactly $n$. A complete proof is in Appendix B. $\square$

Next, we turn our attention to another class of WMI problems, the class $\mathbb{WMI}(\mathbf{\Omega}, \log(n), 2)$, having logarithmic diameter but treewidth 2. This class is also supposed to be "easy" in the sense that it extends the tractable class $\mathbb{WMI}(\mathbf{\Omega}, \log(n), 1)$ by slightly increasing the treewidth by one. Unfortunately, inference in $\mathbb{WMI}(\mathbf{\Omega}, \log(n), 2)$ is also hard.

**Theorem 3.4.** *Let $\mathbb{WMI}(\mathbf{\Omega}, \log(n), 2)$ be the class of WMI problems whose parametric weight function family $\Omega$ satisfies the TWCs. Then inference in $\mathbb{WMI}(\mathbf{\Omega}, \log(n), 2)$ is #P-hard.*

*Sketch of proof.* Analogously to Theorem 3.3, we prove it by constructing a polytime reduction from a #P-complete variant of the subset sum problem to a MI problem whose primal graph has treewidth two but diameter being at most $\log(n)$. A complete proof is provided in Appendix B. $\square$

Note that our result differs from the one presented in [42] for the hardness of the class $2\mathsf{WMI}(\mathbf{\Omega})$, containing WMI problems with SMT formulas being conjunctions of clauses comprising at most two variables. In fact, $\mathbb{WMI}(\mathbf{\Omega}, \log(n), 2)$ is contained in $2\mathsf{WMI}(\mathbf{\Omega})$. As such, we trace the tractablity boundaries of WMI inference with higher precision, as the next corollary states. Its proof follows from Theorems 3.3 and 3.4 and from the sufficiency as demonstrated in Zeng et al. [42].

**Corollary 3.5.** *Let $\mathbb{WMI}(\mathbf{\Omega}, \log(n), t)$ be the class of WMI problems whose parametric weight function family $\Omega$ satisfies the TWCs. Then $\mathbb{WMI}(\mathbf{\Omega}, \log(n), t)$ is a tractable WMI class for inference if-and-only-if treewidth $t = 1$.*

These complexity results set the standard for the solver complexity: every exact WMI solver that aims to be efficient, needs to operate in the regime of Corollary 3.5. However, real-world problems do not always conform to the structural desiderata for primal graphs stated in it. This implies that efficient approximations might not only be useful in these scenarios, but *needed*. In the next section we fill this gap, by introducing our approximate WMI solver that navigates the tractable islands in WMI problems by performing efficient inference on a relaxed version of intractable WMI problems.

# 4   RECOIN: Relax, compensate and then integrate

Our algorithm to approximate WMI inference comprises three phases: i) *RElaxing* an intractable WMI model into a simpler one amenable to exact inference by removing dependencies from it; then ii) introduce certain literals and weights to *COmpensate* for the dependency structure lost in this way and iii) optimize them by solving a series of exact *INtegration* problems. We name it RECOIN. With RECOIN we can navigate *a spectrum of approximations* — with the original primal graph $\mathcal{G}_\Delta$ on one end, and a fully disconnected version on the other — by removing more and more edges. As such, RECOIN can be viewed as extending the *relax-compensate-recover* (RCR) framework [14, 16, 17] for approximate inference on discrete probabilistic models to continuous representations and in presence of algebraic constraints.

## 4.1   Relaxation: introducing and then "breaking" equivalence constraints

The aim of the relaxation step is to obtain a new SMT formula $\Delta^{\mathsf{rel}}$ such that its associated primal graph $\mathcal{G}_{\Delta^{\mathsf{rel}}}$, serves as the simplification of the original $\mathcal{G}_\Delta$ by removing a given set of edges. We will show that the removal of any edge can be formulated as the removal of an equivalence edge [17]. This process consists of two steps. First, we create an *augmented formula* $\Delta^{\mathsf{aug}}$ by introducing new variables to $\Delta$ and enforcing them to act as *copies* of certain original variables by explicitly adding *equivalence constraints*. Second, we deliver the relaxed $\mathcal{G}_{\Delta^{\mathsf{rel}}}$ by removing these equivalence constraints.

**Augmentation.**   The detailed process of distilling a new augmented model $(\Delta^{\mathsf{aug}}, \mathcal{W}^{\mathsf{aug}})$ from $(\Delta, \mathcal{W})$, given a subset of edges $\mathcal{E}_d \subseteq \mathcal{E}$ in $\mathcal{G}_\Delta$ to remove, is listed in Algorithm 2 in Appendix C. At its core, there are routines for copying one variable and adding the corresponding equivalence constraints and compensating literals. For each edge $X_i - X_j \in \mathcal{E}_d$ to be removed, one of its variables is arbitrarily selected, say $X_i$. Then a variable $X_i^c$, as a copy of the chosen $X_i$, is introduced in $\Delta^{\mathsf{aug}}$ as well as one equivalence constraint between the two as the literal $\hat{\ell} : (X_i^c = X_i)$ with associated weight function $\delta(X_i, X_i^c)$ where $\delta$ is the Dirac delta function. Then we properly rename all occurrences of $X_i$ by $X_i^c$ in the literals appearing in the clauses of $\Delta^{\mathsf{aug}}$ that also contain $X_j$ and introduce copied literals for the univariate clauses over $X_i$ only. These steps cause the primal graph $\mathcal{G}_{\Delta^{\mathsf{aug}}}$ to now contain the dependency $X_i - X_i^c - X_j$ but not $X_i - X_j$.

Note that the augmented WMI model $(\Delta^{\mathsf{aug}}, \mathcal{W}^{\mathsf{aug}})$ now contains more variables than the original one. Specifically, for each variable $X_i \in \mathcal{G}_\Delta$ we might have introduced $C_i$ different copies in $\mathcal{G}_{\Delta^{\mathsf{aug}}}$, denoted as $X_i^1, \ldots, X_i^{C_i}$, if we removed $C_i$ edges over $X_i$. We will denote the original $X_i$ as $X_i^0$ for notation consistency. Even if the dimensionality of the augmented WMI problem is increased by augmentation, the next propositions are guaranteeing that we are not altering the partition function and the marginal distributions of $\mathsf{Pr}_\Delta$, and that introducing equivalence constraints does not alter the induced distribution.

**Proposition 4.1.** *Let $\Delta$ be an SMT formula with primal graph $\mathcal{G}_\Delta$ and per-literal weight functions $\mathcal{W}$, and let $\Delta^{\mathsf{aug}}$ and $\mathcal{W}^{\mathsf{aug}}$ be the output of Algorithm 2 when applied to $\Delta$ and $\mathcal{G}_\Delta$ given a certain subset of edges in $\mathcal{G}_\Delta$. Then it holds that $\mathsf{WMI}(\Delta, \mathcal{W}) = \mathsf{WMI}(\Delta^{\mathsf{aug}}, \mathcal{W}^{\mathsf{aug}})$. Moreover, for any $X_i$ in $\mathcal{G}_\Delta$ and univariate literal $\ell$ over $X_i$, it holds that $\mathsf{Pr}_\Delta(\ell) = \mathsf{Pr}_{\Delta^{\mathsf{aug}}}(\ell)$.*

**Removing equivalence constraints.**   Given an augmented model $(\Delta^{\mathsf{aug}}, \mathcal{W}^{\mathsf{aug}})$, we remove equivalence constraints introduced at the augmentation step to obtain the relaxed model $(\Delta^{\mathsf{rel}}, \mathcal{W}^{\mathsf{rel}})$. As a result, each original variable in $\mathcal{G}_{\Delta^{\mathsf{rel}}}$ will be detached from its copies, thus ignoring the dependencies encoded by the edges $\mathcal{E}_d$ that were marked to be removed. Algorithm 3 details this procedure. Note that relaxation "breaks" the augmented formula $\Delta^{\mathsf{aug}}$ into a relaxed part $\Delta^{\mathsf{rel}}$ and a "remaining part" $\Delta^{\mathsf{rem}}$, which contains the equivalence constraints just removed.

**Example 4.2.** *Consider the WMI model $(\Delta, \mathcal{W})$ of Example 2.2. Its augmented formula $\Delta^{\mathsf{aug}}$ obtained by applying Algorithm 2 for edges $\mathcal{E}_d = \{X - Z\}$ to be removed (orange), and its relaxed formula $\Delta^{\mathsf{rel}}$ and remaining formula $\Delta^{\mathsf{rem}}$ obtained by Algorithm 3 have their primal graphs shown on the left, center and right below respectively. The detailed WMI models for each are shown in Appendix A.*

**Which edges to relax?**   After relaxing enough constraints, we can obtain a WMI problem amenable to exact inference, for example, one whose primal graph $\mathcal{G}_{\Delta^{\mathsf{rel}}}$ has treewidth one and logarithmic

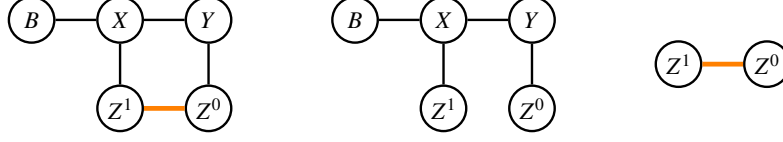

diameter. Running an exact WMI solver on such a problem would already deliver a cheap way to perform approximate inference. However, the quality of such an approximation can be greatly improved if we compensate for the relaxed constraints. We will discuss this in the next section.

A question remains: *how to select the set of edges $\mathcal{E}_d$ to relax?* Note that the more edges we remove from $\Delta$, the easier it is to perform inference on $\Delta^{\text{rel}}$ given fewer dependencies, but the lower the approximation quality, and the harder to compensate for them all, since it would differ from the augmented model more, and meanwhile from the original model as Proposition 4.1 indicates. For example, removing all edges in $\mathcal{G}_\Delta$ will yield a fully disconnected $\mathcal{G}_{\Delta^{\text{rel}}}$ where performing exact inference on each component is going to be embarrassingly parallelizable. This would correpond to perform a loopy version of the MP-WMI algorithm. Analogous to its discrete counterpart, loopy belief propagation, it would be susceptible to poor converge rates [31, 14]. Therefore we propose a simple strategy for selecting the edges to be removed, which is to retrieve a spanning tree of the original primal graph. In Section 6 we demonstrate its practical effectiveness on a range of inference problems of increasing complexity. Devising and evaluating alternative relaxing strategies is an interesting topic for future work.

### 4.2 Compensation

The aim of the compensation phase is to recover the relaxed equivalence constraints and hence, make the distribution $\text{Pr}_{\Delta^{\text{rel}}}$ better approximate $\text{Pr}_{\Delta^{\text{aug}}}$ and thus better approximate $\text{Pr}_\Delta$ as Proposition 4.1 suggests. In order to do so, we introduce new literals, named *compensating literals*, to the variables and their copies in the relaxed formula $\Delta^{\text{rel}}$ and equip them with parameterized weights, named *compensating weights*, and further we optimize them in order to synchronize the variable marginals among a copied variable and its copies.

For each variable $X_i = X_i^0$ and its $C_i$ copies $X_i^1, \ldots, X_i^{C_i}$ in formula $\Delta^{\text{rel}}$, we generate $K$ different univariate literals of the form $\ell_{i,k}^c : (X_i^{(c)} \leq \sigma_{i,k} \cdot \tau_{i,k})$ for $k = 1, \ldots, K$ and $c = 0, 1, \ldots, C_i$ where each $\sigma_{i,k}$ and $\tau_{i,k}$ are respectively drawn at uniform from $\{+1, -1\}$ and the support of $X_i$ as encoded in $\Delta_i^{\text{rel}}$. Note that the $\sigma_{i,k}, \tau_{i,k}$ are *shared* across all the copies. Algorithm 4 in Appendix C summarizes this procedure. Each compensating literal $\ell_{i,k}^c$ is therefore responsible for a portion of the support of the marginal distribution of $X_i^c$, and also for the (unnormalized) marginal density of $X_i^c$ by equipping it with a parameterized weight $w_{\ell_{i,k}^c}$.

To retain tractable inference, the parametric function family chosen for each $w_{\ell_{i,k}^c}$ should satisfy the TWCs as discussed in section 3. Striving for simplicity, we employ constant weights of the form $w_{\ell_{i,k}^c} := \exp(\theta_{i,k}^c)$. Therefore, our induced marginal density takes the form of a piecewise constant approximation. As such, by increasing the number of compensating literals $K$ one could obtain a finer approximation, however at the price of introducing more parameters to optimize for. We empirically investigate the effect of increasing $K$ in our experiments in section 6.

### 4.3 Iterative integration

Instead of matching marginal density functions we settle for the weaker condition of *matching the marginal probabilities of the newly introduced compensating literals*. This in turn can be stated by the following set of equivalence constraints for each variable $X_i$:

$$\text{Pr}_{\Delta^{\text{rem}}}\left(\bigwedge_{c=0}^{C_i} \ell_{k,i}^c\right) = \text{Pr}_{\Delta^{\text{rel}}}(\ell_{k,i}^0) = \text{Pr}_{\Delta^{\text{rel}}}(\ell_{k,i}^1) = \cdots = \text{Pr}_{\Delta^{\text{rel}}}(\ell_{k,i}^{C_i}), \; for \; k = 1, \cdots, K. \quad (2)$$

where the first term $\text{Pr}_{\Delta^{\text{rem}}}\left(\bigwedge_{j=0}^{C_i} \ell_{k,i}^c\right)$ is the probability of the compensating literals in the remaining WMI model $(\Delta^{\text{rem}}, \mathcal{W}^{\text{rem}})$ and $\text{Pr}_{\Delta^{\text{rel}}}(\ell_{k,i}^c)$ are the probabilities of compensating literals in the relaxed formula $\Delta^{\text{rel}}$. Intuitively, for a single equivalence constraint that has been relaxed, there exists a set of

---

**Algorithm 1** ReCoIn $(\Delta, \mathcal{W}, K)$

---

**Input:** a WMI model $(\Delta, \mathcal{W})$, $K$ number of compensating literals
**Output:** $(\Delta^{\mathsf{rel}}, \mathcal{W}^{\mathsf{rel}})$: a relaxed and compensated WMI model

1: $\mathcal{E}_d \leftarrow \mathsf{initStrategy}(\Delta, \mathcal{W})$                $\triangleright$ Select edges to remove
2: $\Delta^{\mathsf{aug}}, \mathcal{W}^{\mathsf{aug}}, \mathcal{L} \leftarrow \mathsf{augmentModel}(\Delta, \mathcal{W}, \mathcal{E}_d)$
3: $(\Delta^{\mathsf{rel}}, \mathcal{W}^{\mathsf{rel}}), (\Delta^{\mathsf{rem}}, \mathcal{W}^{\mathsf{rem}}) \leftarrow \mathsf{relaxModel}(\Delta^{\mathsf{aug}}, \mathcal{W}^{\mathsf{aug}}, \mathcal{L})$
4: $\Delta^{\mathsf{rel}}, \mathcal{W}^{\mathsf{rel}} \leftarrow \mathsf{addingCompensations}(\Delta^{\mathsf{rel}}, \mathcal{W}^{\mathsf{rel}}, \mathcal{L}, K)$
5: **while** *not* converged **do**
6:      **for** $X_i \in \mathsf{copiedNodes}(\Delta^{\mathsf{rel}})$ **do**
7:          **for** $k = 1, \ldots, K$ **do**
8:              $r^k \leftarrow \mathsf{WMI}(\Delta^{\mathsf{rem}}, \mathcal{W}^{\mathsf{rem}}) \,/\, \mathsf{WMI}(\Delta^{\mathsf{rem}} \wedge \bigwedge_{c=0}^{C_i} \ell_{k,i}^c, \mathcal{W}^{\mathsf{rem}}) - 1$
9:              **for** $c = 0, 1, \ldots, C_i$ **do**
10:                  $\theta_{k,i}^{c,(t+1)} \leftarrow \log(r^k \alpha_{k,\sigma(c)}) - \log(1 - \alpha_{k,\sigma(c)}) - \sum_{c' \neq c} \theta_{k,i}^{c,(t)}$
11: **Return** $(\Delta^{\mathsf{rel}}, \mathcal{W}^{\mathsf{rel}})$

---

parameters $\theta$ for the compensating weights that exactly match the probabilities in Equation 2 and hence guarantee exact marginal recovery [14]. The next theorem better formalizes it.

**Theorem 4.3.** *Suppose that a relaxed model* $(\Delta^{\mathsf{rel}}, \mathcal{W}^{\mathsf{rel}})$ *and a remaining model* $(\Delta^{\mathsf{rem}}, \mathcal{W}^{\mathsf{rem}})$ *are obtained by relaxing a single equivalence constraint* $(X_i = X_i^c)$ *from an augmented model* $\Delta^{\mathsf{aug}}$, *and that the primal graph of* $\Delta^{\mathsf{rel}}$ *is split into two disconnected components by the relaxation. Let* $(\ell_{i,k}, \ell_{i,k}^c)$ *for* $k = 1, \ldots, K$ *be the K pairs of compensating literals introduced, and* $\theta_{k,i}, \theta_{k,i}^c$, *for* $k = 1, \ldots, K$, *be the parameters attached to the compensating weights. Then Equation 2 holds when the compensating weight parameters satisfy the following equalities.*

$$\theta_{k,i} = \log \frac{r^k \alpha_{k,c}}{1 - \alpha_{k,c}} - \theta_{k,i}^c, \quad \theta_{k,i}^c = \log \frac{r^k \alpha_k}{1 - \alpha_k} - \theta_{k,i} \quad for \;\; k = 1, \ldots, K \tag{3}$$

*where*

$$r^k = \frac{\mathsf{WMI}(\Delta^{\mathsf{rem}} \wedge \neg\ell_{k,i} \wedge \neg\ell_{k,i}^c, \mathcal{W}^{\mathsf{rem}})}{\mathsf{WMI}(\Delta^{\mathsf{rem}} \wedge \ell_{k,i} \wedge \ell_{k,i}^c, \mathcal{W}^{\mathsf{rem}})}, \;\; \alpha_k = \mathsf{Pr}_{\Delta^{\mathsf{rel}}}(\ell_{i,k}), \; \alpha_{k,c} = \mathsf{Pr}_{\Delta^{\mathsf{rel}}}(\ell_{i,k}^c), \; for \; k = 1, \ldots, K. \tag{4}$$

Theorem 4.3 suggests an iterative optimization scheme to find the fixed point solutions for all the compensating parameters introduced to compensate multiple relaxed equivalence constraints. Specifically, starting from a random initialization of the parameters of the compensating weights,[3] at each iteration $t + 1$, we can update each parameter $\theta_{k,i}^{c,(t+1)}$ as

$$\theta_{k,i}^{c,(t+1)} \leftarrow \log(r^k \alpha_{k,\pi(c)}) - \log(1 - \alpha_{k,\pi(c)}) - \sum_{c' \neq c} \theta_{k,i}^{c',(t)}, \tag{5}$$

where $\pi$ is a permutation over the copies and each $\alpha_{k,\pi(c)}$ is computed as the probability of $\ell_{k,i}^{\pi(c)}$ according to the relaxed model.

Therefore, at each iteration $t$, we need to solve $2K$ integration problems for computing the $r^k$ terms and $C_i \cdot K$ integrations for $\mathsf{Pr}_{\Delta^{\mathsf{rel}}}(\ell_{k,i}^{\pi(c)})$ for each pair of variable and its copies. While in principle we could use any exact WMI solver to solve these problems, we adopt MP-WMI [42] because it is the fastest solver yet for tree-shaped and bounded diameter problems, and even more importantly, it allows to *amortize inference across queries*. That is, we can compute all the $C_i \cdot K$ literal probabilities in a single message-passing step with it.

From this perspective, ReCoIn generates a sequence of induced distributions $\mathsf{Pr}_{\Delta^{\mathsf{rel}}}^{(1)}, \ldots, \mathsf{Pr}_{\Delta^{\mathsf{rel}}}^{(2)}, \mathsf{Pr}_{\Delta^{\mathsf{rel}}}^{(t)}$, that should converge to a fixed-point distribution. In practice to check for convergence, one can monitor the quality of the literal probability approximations and stop when a threshold $\epsilon$ is met before

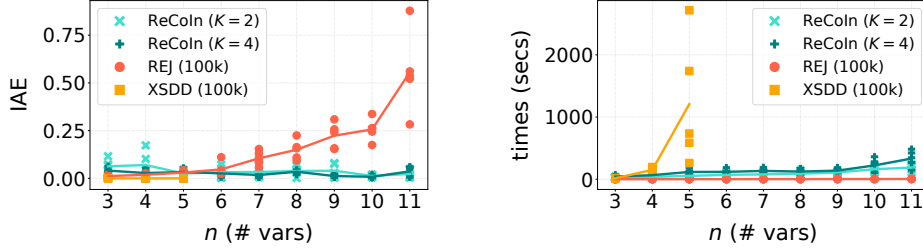

Figure 1: Average integrated absolute errors (left) and times in seconds (right) for 5 problems of increasing size ($n$, x-axis) for ReCoIn and competitors. Number of compensating literals (2-4) or samples used are in parentheses. Mean values per problem size are connected by a line.

a certain number of iterations are done. We choose the threshold to be the maximum $L\text{-}\infty$ norm of compensation literal probability differences. To ease convergence, we apply *dampening*, that is, we smooth each parameter update at iteration $t+1$ by a factor $\lambda > 0$: $\theta_{k,i}^{c,(t+1)} \leftarrow (1-\lambda)\cdot\theta_{k,i}^{c,(t+1)} + \lambda\cdot\theta_{k,i}^{c,(t+1)}$. This completes the steps in our ReCoIn solver. Algorithm 1 recaps them.

## 5   Related Work

The RCR framework has been particularized for approximating marginals [14, 16, 18], partition functions [17], and for maximization [15] or lifted inference scenarios [37], but always for discrete variables. ReCoIn is the first extension to hybrid domains with SMT($\mathcal{LRA}$) algebraic constraints.

Among the exact WMI solvers, the majority ignores the problem structure to be as general-purpose as possible [8, 34, 35, 29]. However, by doing so they are unable to scale beyond tens of variables in practice. Conversely, recent efficient alternatives such as SMI [41] and MP-WMI [42] can greatly scale but only on WMI problems amenable to tractable inference (cf. Section 3). We leverage the strengths of the latter to efficiently solve iterative integration problems in ReCoIn.

So far, most approximate WMI solvers rely on sampling, and as such inherit all the classical issues of Monte Carlo approaches like poor scalability and convergence [19]. Among these, SAMPO [43] employs Gibbs sampling but does not support generic polynomial weights. A very recent alternative is a fully polynomial randomized approximation scheme [1]. However, it can only operate on DNF SMT formulas, and it is not applicable to our CNF representation as a conversion into DNF can blow up the problem size. Other MCMC variants [3, 2, 4] operating with algebraic constraints, while more effective, cannot be readily used for WMI inference problems. The only alternative to sampling schemes is the hashing-based WMI algorithm [9] which is known to perform poorly on non-trivial problems due the hardness of calibrating the *tilt* [10].

In the next section we compare against the fastest baseline available, the rejection sampler implemented in the pywmi library [30] and a more advanced variant of rejection sampling that greatly increases the acceptance rate of the rejection sampler by compiling an SMT formula into an XSDD structure [44].

## 6   Experiments

We aim to answer the following questions: **(Q1)** how fast and scalable is ReCoIn?, **(Q2)** how accurate are its approximations?, **(Q3)** what is the effect of increasing the number of compensating literals $K$?

We generate WMI problems whose primal graphs are random Watts-Strogatz graphs [39] with increasing size $n = 1, \ldots, 11$, with two additional neighbor connections and probability of rewiring 0.5, to which we attach randomly generated clauses of length 2 and piecewise constant densities. For each setting we generate 5 independent problems.

We run ReCoIn for up to 20 iterations, employing a dampening coefficient $\lambda = 0.5$ in two settings that differ by the number of compensating literals $K = 2, 4$. We compare it against the fastest sampling scheme available, the rejection sampler (REJ) implemented in [30] and the hybrid solver XSDD(Sampling) [44] that employs sophisticated knowledge-compilation [21] techniques [29] to guide sampling. For both REJ and XSDD we employ 100 thousand samples per query.

To compare the quality of approximations for a problem, we compute for a model $\mathcal{M}$ the mean integral absolute error (IAE) as $\frac{1}{|\mathbf{X}|} \sum_{i=1}^{|\mathbf{X}|} \sum_{j=1}^{B} |\mathsf{Pr}_G(X_i \in b_j) - \mathsf{Pr}_M(X_i \in b_j)|$ where we partition the support for each marginal $i = 1, \ldots, n$ into $B$ equal-widths bins $b_j$ for $j = 1, \ldots, B$ and compare the probability $\mathsf{Pr}_M$ according to model $\mathsf{M}$ against the ground truth $\mathsf{Pr}_G$, which we compute using PA [34]. We employ PA as it is so far the most reliable general-purpose exact WMI solver [42]. Note that as such REJ and XSDD are bounded to solve $|\mathbf{X}| \cdot B$ independent WMI problems, while RECOIN can naturally amortize $|\mathbf{X}| \cdot B$ queries after a single run of optimization (cf. Section 4.3). We impose a timeout of 1 hour.

Figure 1 reports the IAEs and running times (in seconds) for all problems, settings and competitors. Concerning **Q1** and **Q2**, RECOIN is the best performer overall. The naive sampling strategy in REJ, while being the fastest as expected, cannot exploit the structure in the problem and clearly suffers from the curse of dimensionality. Conversely, XSDD can deliver accurate approximations thanks to compiling the problem structure, but on highly loopy graphs compilation cannot scale beyond $n = 5$. On the other hand, RECOIN gracefully scales to larger problem sizes and multiple queries, and delivers very low IAE scores that are close to the best by XSDD on small problem sizes. Note that while RECOIN can solve much larger problems within our timeout, we could not retrieve a ground truth for them with PA in reasonable time (more than 24 hours per problem).

Concerning **Q3**, more compensating literals ($K = 4$) are achieving marginally lower IAEs at the expense of linearly increasing running times. Exploring the time-accuracy trade-off by increasing $K$ or employing different relaxation strategies is an interesting avenue to investigate in the future. All in all, this empirical evidence candidates RECOIN as one of the best general-purpose approximate WMI solvers in the current landscape of WMI solvers.

# 7  Conclusions

In this work we advanced the WMI framework by tracing the theoretical requirements for tractable WMI inference with the highest precision so far. We introduced RECOIN as the first solver that by exploiting our tractability insights can reliably scale approximate inference on general WMI problems. We believe these two contributions can help strengthen our theoretical understanding on the challenges and guarantees around approximate hybrid probabilistic inference and at the same time propel the construction of more efficient and scalable WMI solvers.

## Acknowledgement

The authors would like to thank Arthur Choi for several insightful discussions about the RCR framework. This work is partially supported by NSF grants #IIS-1943641, #IIS-1633857,#CCF-1837129, DARPA grant #N66001-17-2-4032, a Sloan Fellowship, Intel, and Facebook. This work has received funding from the European Research Council (ERC) under the European Union's Horizon 2020 research and innovation programme (grant agreement No. [694980] SYNTH: Synthesising Inductive Data Models).

## Broader Impact

Our contributions in this work can be filed under the label of basic research in probabilistic inference. As a work of basic research it might have a very broad impact. Therefore it is hard to imagine specific negative outcomes at this stage. Concerning benefits, on the other hand, our complexity results will help the community working on probabilistic inference on hybrid domain at large as they lay the foundation for more theoretical research. On the other hand, our general-purpose approximate WMI inference scheme could be particularized by other researchers to fit specific application scenarios. It is hard to foresee or restrict the range of these possible applications. We note that WMI and SMT technologies have been previously used in probabilistic programming and program verification, two very vast fields on their own. Lastly, we are focusing on and advancing inference per se, therefore there is no specific learning phase, or data involved. Our solver is going to perform inference over the distribution induced over an arbitrary SMT theory given as input, if such a theory encodes bias in some form, this bias will clearly be reflected in the probabilistic queries the users are going to ask.

## Footnotes

[3]Following Choi and Darwiche [16], we initialize all parameters to 1.

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
