[Supplementary Material]

# A  Examples

## A.1  Reduction to WMI models on continuous variables only

In this section, we show one example of the polytime reduction from a WMI model with continuos and discrete ones into one over continuous variables only, as introduced in [41].

**Example A.1** (Reduction From WMI to $\text{WMI}_\mathbb{R}$). *Consider the WMI model $(\Delta, \mathcal{W})$ where $\Delta$ is the SMT formula over continuous variables $X, Y, Z$ and Boolean variable $B$ as introduced in Example 2.2 with the per-literal weights $\mathcal{W}$ as introduced in Example 2.5. Then the WMI model $(\Delta', \mathcal{W}')$ over continuous variables only $X, Y, Z, T_B$, where $T_B$ is a freshly introduced continuous variable, obtained by the reduction of Zeng and Van den Broeck [41] is shown below.*

$$\Delta' = \begin{cases} 0 \le X \le 2 \wedge 1 \le Y \le 2 \wedge 0 \le Z \le 2 \\ X \ge 1 \vee (-1 \le T_B \le 1) \\ X + Y \le 3 \wedge X + Z \ge 2 \wedge Y + Z \le 3 \end{cases}$$

*where $\mathcal{W}' = \{w_{\ell_1}(T_B) := 2;\ w_{\ell_2}(x) := x^2;\ w_{\ell_3}(y, z) := 2yz;\ w_{\ell_4}(x, y) := x + y\}$ where $\ell_1 := 0 \le T_B \le 1$, $\ell_2 := x \ge 1$, $\ell_3 := y + z \le 3$, $\ell_4 := x + y \le 3$ and all the weights associated to other literals are constantly 1 except $\neg \ell_2$ which is 0.*

*Note that the primal graph $\mathcal{G}_{\Delta'}$ (above, right) is isomorphic to the primal graph $\mathcal{G}_\Delta$ and that the weighted model integral of model $(\Delta', \mathcal{W}')$ is left unchanged:*

$$\text{WMI}(\Delta', \mathcal{W}'; X, Y, Z, T_B) = \int_{-1}^{0} dt_B \int_{1}^{2} dx \int_{1}^{-x+3} dy \int_{-x+2}^{-y+3} x^2 \cdot 1 \cdot (x + y) \cdot 2yz\ dz +$$

$$+ \int_{0}^{1} dt_B \int_{1}^{2} dx \int_{1}^{-x+3} dy \int_{-x+2}^{-y+3} x^2 \cdot 2 \cdot (x + y) \cdot 2yz\ dz = \frac{11173}{480} = \text{WMI}(\Delta, \mathcal{W}; X, Y, Z, B).$$

*then we will denote the integrands as $u_1(x, y, z) = x^2 \cdot 1 \cdot (x + y) \cdot 2yz,\ u_2(x, y, z) = x^2 \cdot 2 \cdot (x + y) \cdot 2yz$.*

## A.2  RECOIN steps: from augmentation to relaxation

Here we complete Example 4.2 by providing the weight functions associated to the WMI models RECOIN operates on.

**Example A.2** (Augmentation). *Consider the WMI model $(\Delta', \mathcal{W}')$ over continuous variables $X, Y, Z, T_B$ as introduced in Example A.1. Given the edges to remove $\mathcal{E}_d = \{X - Z\}$, the augmented WMI model $(\Delta^{\text{aug}}, \mathcal{W}^{\text{aug}})$ over variables $X, Y, Z = Z^0, Z^1, T_B$ as obtained from Algorithm 2 is represented below.*

$$\Delta^{\text{aug}} = \begin{cases} 0 \le X \le 2 \wedge 1 \le Y \le 2 \\ 0 \le Z^0 \le 2 \wedge 0 \le Z^1 \le 2 \\ -1 \le T_B \le 1 \\ X \ge 1 \vee T_B > 0 \\ X + Y \le 3 \wedge X + Z^1 \ge 2 \wedge Y + Z^0 \le 3 \\ Z^0 = Z^1 \end{cases}$$

*and $\mathcal{W}^{\text{aug}} = \{w_{\ell_1}(T_B) := 2;\ w_{\ell_2}(x) := x^2;\ w_{\ell_3}(y, z^0) := 2yz^0;\ w_{\ell_4}(x, y) := x + y;\ w_{\ell_5}(z^0, z^1) := \delta(z^0, z^1)\}$ where $\ell_1 := 0 \le T_B$, $\ell_2 := x \ge 1$, $\ell_3 := y + z^0 \le 3$, $\ell_4 := x + y \le 3$, $\ell_5 := Z^0 = Z^1$ and all the weights associated to other literals are constantly 1 except $\neg \ell_2$ which is 0.*

*Note that the weighted model integral of model $(\Delta^{\text{aug}}, \mathcal{W}^{\text{aug}})$ is unchanged as below:*

$$\text{WMI}(\Delta^{\text{aug}}, \mathcal{W}^{\text{aug}}; X, Y, Z^0, Z^1, T_B) =$$

$$= \int_{-1}^{0} dt_B \int_{1}^{2} dx \int_{1}^{-x+3} dy \int_{0}^{-y+3} \int_{-x+2}^{1} x^2 \cdot (2 + 1) \cdot (x + y) \cdot 2yz^0 \delta(z_0 - z_1) dz^1 dz^0$$

$$= \int_{1}^{2} dx \int_{1}^{-x+3} dy \int_{-x+2}^{-y+3} x^2 \cdot (2 + 1) \cdot (x + y) \cdot 2yz^0\ dz^0$$

$$= \frac{11173}{480} = \mathsf{WMI}(\Delta', \mathcal{W}'; X, Y, Z, T_B) = \mathsf{WMI}(\Delta, \mathcal{W}; X, Y, Z, B).$$

*Further we will show in Proof B.3 that generally the WMI of the augmented model remains unchanged.*

**Example A.3** (Relaxation). *Consider the augmented WMI model* $(\Delta^{\mathsf{aug}}, \mathcal{W}^{\mathsf{aug}})$ *over continuous variables* $X, Y, Z^0, Z^1, T_B$ *as introduced in Example A.2. Given the equivalence constraint to remove* $\{Z^0 = Z^1\}$*, the relaxed WMI model* $(\Delta^{\mathsf{rel}}, \mathcal{W}^{\mathsf{rel}})$ *and its remaining part* $(\Delta^{\mathsf{rem}}, \mathcal{W}^{\mathsf{rem}})$ *as obtained from Algorithm 3 are represented below.*

$$\Delta^{\mathsf{rel}} = \begin{cases} 0 \le X \le 2 \wedge 1 \le Y \le 2 \wedge 0 \le Z^0 \le 2 \wedge 0 \le Z^1 \le 2 \\ X \ge 1 \vee (-1 \le T_B \le 1) \\ X + Y \le 3 \wedge X + Z^1 \ge 2 \wedge Y + Z^0 \le 3 \end{cases}$$

$$\Delta^{\mathsf{rem}} = \begin{cases} 0 \le Z^0 \le 2 \wedge 0 \le Z^1 \le 2 \\ Z^0 = Z^1 \end{cases}$$

*and* $\mathcal{W}^{\mathsf{rel}} = \{w_{\ell_1}(T_B) := 2;\ w_{\ell_2}(x) := x^2;\ w_{\ell_3}(y, z^0) := 2yz^0;\ w_{\ell_4}(x, y) := x + y\},\ \mathcal{W}^{\mathsf{rem}} = \{w_{\ell_5}(z^0, z^1) := \delta(z^0, z^1)\}$*, and all the weights associated to other literals are constantly 1 except* $\neg \ell_2$ *which is 0.*

## B    Proofs

### B.1    THEOREM 3.3

*Proof.* We prove our complexity result by reducing a #P-complete variant of the subset sum problem [24] to an MI problem over an SMT($\mathcal{LRA}$) formula $\Delta$ with tree primal graph whose diameter is $n$. This problem is a counting version of subset sum problem saying that given a set of positive integers $S = \{s_1, s_2, \cdots, s_n\}$, and a positive integer $L$, the goal is to count the number of subsets $S' \subseteq S$ such that the sum of all the integers in the subset $S'$ equals to $L$. Notice that our proof can be applied to rational numbers as well and we assume binary representations for numbers.

First, we reduce the counting subset sum problem in polynomial time to a model integration problem by constructing the following SMT($\mathcal{LRA}$) formula $\Delta$ on real variables $\mathbf{X}$ whose primal graph is shown in Figure 2:

Figure 2: Primal graph $\mathcal{G}_\Delta$ used for the #P-hardness reduction in Theorem 3.3. We construct the corresponding formula $\Delta$ such that $\mathcal{G}_\Delta$ has maximum diameter (it is a chain). We graphically augment graph $\mathcal{G}_\Delta$ by introducing blue nodes to indicate that integers $s_i$ in set $S$ are contained in clauses between two variables.

$$\Delta = \begin{cases} \underbrace{s_1 - \frac{1}{2n} < x_1 < s_1 + \frac{1}{2n}}_{\ell(1,0)} \vee \underbrace{-\frac{1}{2n} < x_1 < \frac{1}{2n}}_{\ell(1,1)} \\ \underbrace{x_{i-1} + s_i - \frac{1}{2n} < x_i < x_{i-1} + s_i + \frac{1}{2n}}_{\ell(i,0)} \vee \underbrace{x_{i-1} - \frac{1}{2n} < x_i < x_{i-1} + \frac{1}{2n}}_{\ell(i,1)}, \quad i = 2, \cdots n \end{cases}$$

For brevity, we denote the first and the second literal in the $i$-th clause by $\ell(i, 0)$ and $\ell(i, 1)$ respectively as shown above. Also We choose two constants $l = L - \frac{1}{2}$ and $u = L + \frac{1}{2}$.

In the following, we prove that $n^n \mathsf{MI}(\Delta \wedge (l < X_n < u))$ equals to the number of subset $S' \subseteq S$ whose element sum equals to $L$, which indicates that WMI problem whose tree primal graph has diameter $\Theta(n)$ is #P-hard.

Let $\boldsymbol{a}^k = (a_1, a_2, \cdots, a_k)$ be some assignment to Boolean variables $(A_1, A_2, \cdots, A_k)$ with $a_i \in \{0, 1\}$, $i \in [k]$. Given an assignment $\boldsymbol{a}^k$, we define subset sums to be $S(\boldsymbol{a}^k) \triangleq \sum_{i=1}^{k} a_i s_i$, and formulas $\Delta_{\boldsymbol{a}^k} \triangleq \bigwedge_{i=1}^{k} \ell(i, a_i)$.

**Claim B.1.** *The model integration for formula $\Delta_{\boldsymbol{a}^k}$ with an given assignment $\boldsymbol{a}^k \in \{0, 1\}^k$ is* $\mathsf{MI}(\Delta_{\boldsymbol{a}^k}) = (\frac{1}{n})^k$. *Moreover, for each variable $X_i$ in $\Delta_{\boldsymbol{a}^k}$, its satisfying assignments consist of the interval $[\sum_{j=1}^{i} a_j s_j - \frac{i}{2n}, \sum_{j=1}^{i} a_j s_j + \frac{i}{2n}]$. Specifically, the satisfying assignments for variable $X_n$ in formula $\Delta_{\boldsymbol{a}^n}$ can be denoted by the interval $[S(\boldsymbol{a}^n) - \frac{1}{2}, S(\boldsymbol{a}^n) + \frac{1}{2}]$.*

*Proof.* (Claim B.1) First we prove that $\mathsf{MI}(\Delta_{\boldsymbol{a}^k}) = (\frac{1}{n})^k$. For brevity, denote $a_i s_i$ by $\hat{s}_i$. By definition of model integration and the fact that the integral is absolutely convergent (since we are integrating a constant function, i.e., one, over finite volume regions), we have the following equation.

$$\mathsf{MI}(\Delta_{\boldsymbol{a}^k}) = \int_{(x_1, \cdots, x_k) \models \Delta_{\boldsymbol{a}^k}} 1 \; dx_1 \cdots dx_k = \int_{\hat{s}_1 - \frac{1}{2n}}^{\hat{s}_1 + \frac{1}{2n}} dx_1 \cdots \int_{x_{k-2} + \hat{s}_{k-1} - \frac{1}{2n}}^{x_{k-2} + \hat{s}_{k-1} + \frac{1}{2n}} dx_{k-1} \int_{x_{k-1} + \hat{s}_k - \frac{1}{2n}}^{x_{k-1} + \hat{s}_k + \frac{1}{2n}} 1 \; dx_k$$

Observe that for the most inner integration over variable $x_k$, the integration result is $\frac{1}{n}$. By doing this iteratively, we have that $\mathsf{MI}(\Delta_{\boldsymbol{a}^k}) = (\frac{1}{n})^k$.

Next we prove that satisfying assignments for variable $X_i$ in formula $\Delta_{\boldsymbol{a}^k}$ is the interval $[\sum_{j=1}^{i} a_j s_j - \frac{i}{2n}, \sum_{j=1}^{i} a_j s_j + \frac{i}{2n}]$ by mathematical induction. For $i = 1$, since $X_1$ is in interval $[a_1 s_1 - \frac{1}{2n}, a_1 s_1 + \frac{1}{2n}]$, the statement holds in this case. Suppose that the statement holds for $i = m$, i.e. variable $X_m$ has its satisfying assignments in interval $[\sum_{j=1}^{m} a_j s_j - \frac{m}{2n}, \sum_{j=1}^{m} a_j s_j + \frac{m}{2n}]$. Since variable $X_{m+1}$ has its satisfying assignments in interval $[X_m + a_{m+1} s_{m+1} - \frac{1}{2n}, X_m + a_{m+1} s_{m+1} + \frac{1}{2n}]$, then its satisfying assignments consist interval $[\sum_{j=1}^{m+1} a_j s_j - \frac{m+1}{2n}, \sum_{j=1}^{m+1} a_j s_j + \frac{m+1}{2n}]$, that is, the statement also holds for $i = m + 1$. Thus the claim holds. $\square$

The above claim shows how to compute the model integration of formula $\Delta_{\boldsymbol{a}^k}$. We will show in the next claim how to compute the model integration of formula $\Delta_{\boldsymbol{a}^n}$ conjoined with a query $l < X_n < u$.

**Claim B.2.** *For each assignment $\boldsymbol{a}^n \in \{0, 1\}^n$, the model integration of formula $\Delta_{\boldsymbol{a}^n} \wedge (l < X_n < u)$ falls into one of the following cases:*

    *i) If $S(\boldsymbol{a}^n) < L$ or $S(\boldsymbol{a}^n) > L$, it holds that $\mathsf{MI}(\Delta_{\boldsymbol{a}^n} \wedge (l < X_n < u)) = 0$.*
    *ii) If $S(\boldsymbol{a}^n) = L$, it holds that $\mathsf{MI}(\Delta_{\boldsymbol{a}^n} \wedge (l < X_n < u)) = (\frac{1}{n})^n$.*

*Proof.* (Claim B.2) From the previous Claim B.1, it is shown that variable $X_n$ has its satisfying assignments in interval $[S(\boldsymbol{a}^n) - \frac{1}{2}, S(\boldsymbol{a}^n) + \frac{1}{2}]$ in formula $\Delta_{\boldsymbol{a}^n}$ for each $\boldsymbol{a}^n \in \{0, 1\}^n$. If $S(\boldsymbol{a}^n) < L$, given that $S(\boldsymbol{a}^n)$ is a sum of positive integers, then it holds that $S(\boldsymbol{a}^n) + \frac{1}{2} \le (L - 1) + \frac{1}{2} = L - \frac{1}{2} = l$ and therefore, $\mathsf{MI}(\Delta_{\boldsymbol{a}^n} \wedge (l < X_n < u)) = 0$; similarly, if $S(\boldsymbol{a}^n) > L$, then it holds that $S(\boldsymbol{a}^n) - \frac{1}{2} \ge u$ and therefore, $\mathsf{MI}(\Delta_{\boldsymbol{a}^n} \wedge (l < X_n < u)) = 0$. If $S(\boldsymbol{a}^n) = L$, by Claim B.1 we have that the satisfying assignment interval is inside the interval $[l, u]$ and thus it holds that $\mathsf{MI}(\Delta_{\boldsymbol{a}^n} \wedge (l < X_n < u)) = \mathsf{MI}(\Delta_{\boldsymbol{a}^n}) = (\frac{1}{n})^n$. $\square$

In the next claim, we show how to compute the model integration of formula $\Delta$ as well as for formula $\Delta$ conjoined with query $l < X_n < u$ based on the already proven Claim B.1 and Claim B.2.

**Claim B.3.** *The following two equations hold:*

    *i) $\mathsf{MI}(\Delta) = \sum_{\boldsymbol{a}^n} \mathsf{MI}(\Delta_{\boldsymbol{a}^n})$.*
    *ii) $\mathsf{MI}(\Delta \wedge (l < X_n < u)) = \sum_{\boldsymbol{a}^n} \mathsf{MI}(\Delta_{\boldsymbol{a}^n} \wedge (l < X_n < u))$.*

*Proof.* (Claim B.3) Observe that for each clause in $\Delta$, literals are mutually exclusive since each $s_i$ is a positive integer. Then we have that formulas $\Delta_{\boldsymbol{a}^n}$ are mutually exclusive and meanwhile $\Delta = \bigvee_{\boldsymbol{a}^n} \Delta_{\boldsymbol{a}^n}$. Thus it holds that $\mathsf{MI}(\Delta) = \sum_{\boldsymbol{a}^n} \mathsf{MI}(\Delta_{\boldsymbol{a}^n})$. Similarly, we have formulas $(\Delta_{\boldsymbol{a}^n} \wedge (l < X_n < u))$'s are mutually exclusive and meanwhile $\Delta \wedge (l < X_n < u) = \bigvee_{\boldsymbol{a}^n} \Delta_{\boldsymbol{a}^n} \wedge (l < X_n < u)$. Thus the second equation holds. $\square$

Figure 3: Primal graph used for #P-hardness reduction in Theorem 7. We also put blue nodes to indicate that integer $s_i$'s in set $S$ are contained in some clauses and that model integration over some cliques is the sum of some $s_i$'s.

From the above claims, we can conclude that $\mathsf{MI}(\Delta \wedge (l < X_n < u)) = t(\frac{1}{n})^n$ where $t$ is the number of assignments $\boldsymbol{a}^n$ s.t. $S(\boldsymbol{a}^n) = L$. Notice that for each $\boldsymbol{a}^n \in \{0, 1\}^n$, there is a one-to-one correspondance to a subset $S' \subseteq S$ by defining $\boldsymbol{a}^n$ as $a_i = 1$ if and only if $s_i \in S'$; and $S(\boldsymbol{a}^n)$ equals to $L$ if and only if the sum of elements in $S'$ is $L$. Therefore $n^n\mathsf{MI}(\Delta \wedge (l < X_n < u))$ equals to the number of subset $S' \subseteq S$ whose element sum equals to $L$. This finishes the proof for the statement that inference in $\mathcal{WMI}(\boldsymbol{\Omega}, n, 1)$ is #P-hard.

$\square$

## B.2 THEOREM 3.4

*Proof.* Again we prove our complexity result by reducing the #P-complete variant of the subset sum problem [24] to an MI problem over an SMT($\mathcal{LRA}$) formula $\Delta$ with primal graph whose diameter is $\Theta(\log n)$ and treewidth two. In the #P-complete subset sum problem, we are given a set of positive integers $S = \{s_1, s_2, \cdots, s_n\}$, and a positive integer $L$. Notice that our proof can be applied to rational numbers as well and we assume binary representations for numbers. The goal is to count the number of subsets $S' \subseteq S$ such that the sum of all the integers in $S'$ equals $L$.

First, we reduce this problem in polynomial time to a model integration problem with the following SMT($\mathcal{LRA}$) formula $\Delta$ where variables are real and $u$ and $l$ are two constants. Its primal graph is shown in Figure 3. Consider $n = 2^k$, $n, k \in \mathbb{N}$.

$$\Delta = \bigwedge_{i \in [n]} \left( -\frac{1}{4n} < X_{k+1,i} < \frac{1}{4n} \vee -\frac{1}{4n} + s_i < X_{k+1,i} < \frac{1}{4n} + s_i \right) \bigwedge \Delta_t$$

$$\text{where } \Delta_t = \bigwedge_{j \in [k], i \in [2^j]} -\frac{1}{4n} + X_{j+1,2i-1} + X_{j+1,2i} < X_{j,i} < \frac{1}{4n} + X_{j+1,2i-1} + X_{j+1,2i}$$

For brevity, we denote all the variables by $\mathbf{X}$ and denote the literal $-\frac{1}{4n} < X_{k+1,i} < \frac{1}{4n}$ by $\ell(i, 0)$ and literal $-\frac{1}{4n} + s_i < X_{k+1,i} < \frac{1}{4n} + s_i$ by $\ell(i, 1)$ respectively. Also We choose two constants $l = L - \frac{1}{2}$ and $u = L + \frac{1}{2}$. In the following, we prove that $(2n)^{2n-1}\mathsf{MI}(\Delta \wedge (l < X_{1,1} < u))$ equals to the number of subset $S' \subseteq S$ whose element sum equals to $L$, which indicates that model integration problem with primal graph whose diameter is $\Theta(\log n)$ and treewidth two is #P-hard.

Let $\boldsymbol{a}^n = (a_1, a_2, \cdots, a_n) \in \{0, 1\}^n$ be some assignment to Boolean variables $(A_1, A_2, \cdots, A_n)$. Given an assignment $\boldsymbol{a}^n$, define the sum as $S(\boldsymbol{a}^n) \triangleq \sum_{i=1}^n a_i s_i$, and formula as $\Delta_{\boldsymbol{a}^n} \triangleq \bigwedge_{i=1}^n \ell(i, a_i) \wedge \Delta_t$.

**Claim B.4.** *The model integration for formula $\Delta_{a^n}$ with given $a^n \in \{0,1\}^n$ is $\text{MI}(\Delta_{a^n}) = (\frac{1}{2n})^{2n-1}$. Moreover, for each variable $X_{j,i}$ in formula $\Delta_{a^n}$, its satisfying assignments consist of the interval $[\sum_l a_l s_l - \frac{2^{k-j+2}-1}{4n}, \sum_l a_l s_l + \frac{2^{k-j+2}-1}{4n}]$ where $l \in \{l \mid X_{k+1,l}$ is a descendant of $X_{j,i}\}$. Specifically, the satisfying assignments for the root variable $X_{1,1}$ can be denoted the interval $[S(a^n) - \frac{2n-1}{4n}, S(a^n) + \frac{2n-1}{4n}] \subset [S(a^n) - \frac{1}{2}, S(a^n) + \frac{1}{2}]$.*

*Proof.* (Claim B.4) First we prove that $\text{MI}(\Delta_{a^n}) = (\frac{1}{2n})^{2n-1}$. For brevity, denote $a_i s_i$ by $\hat{s}_i$. By definition of model integration and the fact that the integral is absolutely convergent (since we are integrating a constant function, i.e., one, over finite volume regions), we have the following equations

$$\text{MI}(\Delta_{a^n}) = \int_{x \models \Delta_{a^n}} 1 \, d\mathbf{X}$$

$$= \int_{-\frac{1}{4n}+\hat{s}_n}^{\frac{1}{4n}+\hat{s}_n} dx_{k+1,n} \cdots \int_{-\frac{1}{4n}+\hat{s}_1}^{\frac{1}{4n}+\hat{s}_1} dx_{k+1,1} \int_{-\frac{1}{4n}+x_{k+1,n-1}+x_{k+1,n}}^{\frac{1}{4n}+x_{k+1,n-1}+x_{k+1,n}} dx_{k,2^{k-1}} \cdots \int_{-\frac{1}{4n}+x_{2,1}+x_{2,2}}^{\frac{1}{4n}+x_{2,1}+x_{2,2}} 1 \, dx_{1,1} .$$

Observe that for the most inner integration over variable $x_{1,1}$, the integration result is $\frac{1}{2n}$. By doing this iteratively, we have that $\text{MI}(\Delta_{a^k}) = (\frac{1}{2n})^{2n-1}$ where the $2n-1$ comes from the number of variables.

Then we prove that satisfying assignments for variable $X_{j,i}$ in formula $\Delta_{a^n}$ lie in the interval $[\sum_l a_l s_l - \frac{2^{k-j+2}-1}{4n}, \sum_l a_l s_l + \frac{2^{k-j+2}-1}{4n}]$ where $l \in \{l \mid X_{k+1,l}$ is a descendant of $X_{j,i}\}$ by performing mathematical induction in a bottom-up way.

For $j = 1$, any variable $X_{k+2-j,i}$ with $i \in [2^{k+2-j}]$ has satisfying assignments consisting of the interval $[a_i s_i - \frac{1}{4n}, a_i s_i + \frac{1}{4n}]$. Thus the statement holds for this case.

Suppose that the statement holds for $j = m$, that is, for any $i \in [2^{k+2-m}]$, any variable $X_{k+2-m,i}$ has satisfying assignments consisting interval $[\sum_l a_l s_l - \frac{2^m-1}{4n}, \sum_l a_l s_l + \frac{2^m-1}{4n}]$ where $l \in \{l \mid X_{k+1,l}$ is a descendant of $X_{k+2-m,i}\}$.

Then for $j = m+1$ and any $i \in [2^{k+1-m}]$, the variable $X_{k+1-m,i}$ has two descendants, variable $X_{k+2-m,2i-1}$ and variable $X_{k+2-m,2i}$. Moreover, we have that $-\frac{1}{4n} + X_{k+2-m,2i-1} + X_{k+2-m,2i} < X_{k+1-m,i} < \frac{1}{4n} + X_{k+2-m,2i-1} + X_{k+2-m,2i}$. Then the lower bound of the interval for variable $X_{k+1-m,i}$ is $-\frac{1}{4n} + \sum_l a_l s_l - 2\frac{2^m-1}{4n} = \sum_l a_l s_l - \frac{2^{m+1}-1}{4n}$; similarly the upper bound of the interval is $\sum_l a_l s_l + \frac{2^{m+1}-1}{4n}$, where $l \in \{l \mid X_{k+1,l}$ is a descendant of $X_{k+1-m,i}\}$. That is, the statement also holds for $j = m+1$ which finishes our proof. $\square$

The above claim shows what the model integration of formula $\Delta_{a^k}$ is like. We'll show in the next claim what the model integration of formula $\Delta_{a^n}$ conjoined with a query $l < X_{1,1} < u$ is like.

**Claim B.5.** *For each assignments $a^n \in \{0,1\}^n$, the model integration of $\Delta_{a^n} \wedge (l < X_{1,1} < u)$ falls into one of the following cases:*

    *i) If $S(a^n) < L$ or $S(a^n) > L$, then $\text{MI}(\Delta_{a^n} \wedge (l < X_{1,1} < u)) = 0$.*

    *ii) If $S(a^n) = L$, then $\text{MI}(\Delta_{a^n} \wedge (l < X_{1,1} < u)) = (\frac{1}{2n})^{2n-1}$.*

*Proof.* (Claim B.5) From previous Claim B.4, it is shown that variable $X_{1,1}$ has its satisfying assignments in the interval $[S(a^n) - \frac{2n-1}{4n}, S(a^n) + \frac{2n-1}{4n}]$ in formula $\Delta_{a^n}$ for each $a^n \in \{0,1\}^n$.

If $S(a^n) < L$, given that $S(a^n)$ is a sum of positive integers, then it holds that $S(a^n) + \frac{1}{2} \leq (L-1) + \frac{2n-1}{4n} < L - \frac{1}{2} = l$ and therefore, $\text{MI}(\Delta_{a^n} \wedge (l < X_{1,1} < u)) = 0$; similarly, if $S(a^n) > L$, then it holds that $S(a^n) - \frac{1}{2} > u$ and therefore, $\text{MI}(\Delta_{a^n} \wedge (l < X_{1,1} < u)) = 0$. If $S(a^n) = L$, then by Claim B.4 we have that the satisfying assignment interval is inside the interval $[l, u]$ and thus it holds that $\text{MI}(\Delta_{a^n} \wedge (l < X_{1,1} < u)) = \text{MI}(\Delta_{a^n}) = (\frac{1}{2n})^{2n-1}$. $\square$

**Claim B.6.** *The following two equations hold:*

    *i) $\text{MI}(\Delta) = \sum_{a^n} \text{MI}(\Delta_{a^n})$.*

    *ii) $\text{MI}(\Delta \wedge (l < X_{1,1} < u)) = \sum_{a^n} MI(\Delta_{a^n} \wedge (l < X_{1,1} < u))$.*

*Proof.* (Claim B.6) Observe that for each pair of literals $\ell(i, 0)$ and $\ell(i, 1), i \in [n]$, literals are mutually exclusive since each $s_i$ is a positive integer. Then we have that formulas $\Delta_{\boldsymbol{a}^n}$ are mutually exclusive and meanwhile formula $\Delta = \bigvee_{\boldsymbol{a}^n} \Delta_{\boldsymbol{a}^n}$. Thus it holds that $\mathsf{MI}(\Delta) = \sum_{\boldsymbol{a}^n} \mathsf{MI}(\Delta_{\boldsymbol{a}^n})$. Similarly, we have formulas $(\Delta_{\boldsymbol{a}^n} \wedge (l < X_{1,1} < u))$'s are mutually exclusive and meanwhile $\Delta \wedge (l < X_{1,1} < u) = \bigvee_{\boldsymbol{a}^n} \Delta_{\boldsymbol{a}^n} \wedge (l < X_{1,1} < u)$. Thus the second equation holds. □

From the above claims, we can conclude that $\mathsf{MI}(\Delta \wedge (l < X_{1,1} < u)) = t(\frac{1}{2n})^{2n-1}$ where $t$ is the number of assignments $\boldsymbol{a}^n$ s.t. $S(\boldsymbol{a}^n) = L$. Notice that for each $\boldsymbol{a}^n \in \{0,1\}^n$, there is a one-to-one correspondence to a subset $S' \subseteq S$ by defining $\boldsymbol{a}^n$ as $a_i = 1$ if and only if $s_i \in S'$; and $S(\boldsymbol{a}^n)$ equals to $L$ if and only if the sum of elements in $S'$ is $L$. Therefore $(2n)^{2n-1} \mathsf{MI}(\Delta \wedge (l < X_{1,1} < u))$ equals to the number of subset $S' \subseteq S$ whose element sum equals to $L$. This finishes the proof for the statement that inference in $\mathcal{WMI}(\boldsymbol{\Omega}, \log(n), 2)$ is #P-hard. □

## B.3 PROPOSITION 4.1

*Proof.* W.l.o.g, consider the case where the augmented WMI model $(\Delta^{\mathsf{aug}}, \mathcal{W}^{\mathsf{aug}})$ is obtained by removing an edge $X_i - X_j$ and inducing the dependency $X_i - X_i^c - X_j$ from the original WMI model $(\Delta, \mathcal{W})$ as shown in Algorithm 2.

Instrumentally to the proof, we introduce the concept of *total truth assignments* of an SMT($\mathcal{LRA}$) formula $\Delta$. A total truth assignment $\mu$ is defined as a partitioning of all true literals in $\mathcal{L}$, the set of all literals in formula $\Delta$, into a set of literals $\mu_\top$ interpreted as true for a certain total configurations of the variables in $\Delta$ and and the complementary set $\mu_\bot$ containing the literals interpreted as false. Let $tta(\Delta)$ be the set of all total truth assignments for formula $\Delta$.

Notice that when operating on continuous variables only, the definition of WMI in Equation 1 can be rewritten in terms of the total truth assignments to $\Delta$ as follows:

$$\mathsf{WMI}(\Delta, \mathcal{W}) = \sum_{\mu \in tta(\Delta)} \int [\![\boldsymbol{x} \models \mu]\!] \prod_{\ell \in \mathcal{L}} w(\boldsymbol{x})^{[\![\boldsymbol{x} \models \ell]\!]} d\boldsymbol{x} := \sum_{\mu \in tta(\Delta)} Z_\mu. \quad (6)$$

Before we prove that the WMI remains unchanged for the augmented model, we need the following claim.

**Claim B.7.** *Let $tta(\Delta)$ and $tta(\Delta^{\mathsf{aug}})$ be the set of total truth assignments of formula $\Delta$ and that of formula $\Delta^{\mathsf{aug}}$ respectively. Then there exists a bijection between $tta(\Delta)$ and $tta(\Delta^{\mathsf{aug}})$.*

*Proof.* The proof is done by explicitly constructing a bijection $f : tta(\Delta) \to tta(\Delta^{\mathsf{aug}})$ which maps $\mu \in tta(\Delta)$ to $\mu' \in tta(\Delta^{\mathsf{aug}})$ in the following way:

    i) for every $\ell \in \Delta_i$, if $\ell \in \mu_\top$, then $\ell \in \mu'_\top$ and $\ell\{X_i : X_i^c\} \in \mu'_\top$; otherwise $\ell \in \mu'_\bot$ and $\ell\{X_i : X_i^c\} \in \mu'_\bot$.
    ii) for every $\ell \in \Delta_{ij}$, if $\ell \in \mu_\top$, then $\ell\{X_i : X_i^c\} \in \mu'_\top$; otherwise $\ell\{X_i : X_i^c\} \in \mu'_\bot$.
    iii) for every $\ell \notin \Delta_i$ and $\ell \notin \Delta_{ij}$, if $\ell \in \mu_\top$, then $\ell \in \mu'_\top$; otherwise $\ell \in \mu'_\bot$.
    iv) finally, by definition, literal $X_i = X_i^c$ is always in set $\mu'_\top$ (otherwise $\mu'$ would not be a satisfying assignment to formula $\Delta^{\mathsf{aug}}$)

where $\Delta_i$ is the sub-formula containing all the univariate clauses in $\Delta$ referring to $X_i$ only and analogously $\Delta_{ij}$ is the sub-formula containing bivariate clauses in $\Delta$ referring to $X_i$ and $X_j$.

First, note that the function $f$ is well-defined since every literal in formula $\Delta^{\mathsf{aug}}$ is assigned to either set $\mu'_\top$ or set $\mu'_\bot$ by the construction of formula $\Delta^{\mathsf{aug}}$ and this uniquely defines a $\mu' \in tta(\Delta^{\mathsf{aug}})$. Second, by construction, if $f(\mu_1) = f(\mu_2)$ for some $\mu_1, \mu_2 \in tta(\Delta)$, the two total truth assignments $\mu_1$ and $\mu_2$ should have the same set of positive literals as well as the same set of negative literals, which means that $\mu_1 = \mu_2$. Thus, the function $f$ is a one-to-one mapping. Moreover, for each $\mu' \in tta(\Delta^{\mathsf{aug}})$, there exists $\mu \in tta(\Delta)$ obtained by substituting the variable $x_i'$ by $X_i$ and deleting literals in $\Delta_i\{X_i : X_i^c\}$ and literal $X_i = X_i^c$, such that $f(\mu) = \mu'$. That is, the function $f$ is also an onto mapping. Overall, the function $f$ is a bijection between $tta(\Delta)$ and $tta(\Delta^{\mathsf{aug}})$. □

From Equation 6, it follows that to prove that $\mathsf{WMI}(\Delta, \mathcal{W}) = \mathsf{WMI}(\Delta^{\mathsf{aug}}, \mathcal{W}^{\mathsf{aug}})$, it suffices to prove that for each $\mu \in tta(\Delta)$, $Z_\mu$, the integration inside summation corresponding to assignment $\mu$, equates

$Z_{f(\mu)}^{\mathsf{aug}}$ inside $\mathsf{WMI}(\Delta^{\mathsf{aug}}$ with function $f$ as defined in Claim B.7. Let $\mathbf{X}_{-i} = \mathbf{X} \setminus \{X_i\}$. Then the set of variables appearing in formula $\Delta^{\mathsf{aug}}$ can be written as $\mathbf{X}_{-i} \cup \{X_i\} \cup \{X_i^c\}$. Let $\Delta_{ij}^{\mathsf{aug}} := \Delta_{ij}\{X_i : X_i^c\}$ and $\Delta^{\mathsf{aug}} := \Delta^{\bar{\mathsf{a}}\mathsf{ug}} \wedge (X_i = X_i^c)$. We explicitly formulate the integration $Z_\mu$ and $Z_{f(\mu)}^{\mathsf{aug}}$ as follows.

$$Z_\mu = \int [\![ \boldsymbol{x} \models \mu ]\!] \prod_{\ell \in \Delta} w_\ell(\boldsymbol{x})^{[\![ \boldsymbol{x} \models \ell ]\!]} d\boldsymbol{x}$$

$$Z_{f(\mu)}^{\mathsf{aug}} = \int [\![ \boldsymbol{x}_{-i}, x_i, x_i^c \models f(\mu) ]\!] \prod_{\ell \in \Delta^{\bar{\mathsf{a}}\mathsf{ug}}} w_\ell(\boldsymbol{x}_{-i}, x_i, x_i^c)^{[\![ \boldsymbol{x}_{-i}, x_i, x_i^c \models \ell ]\!]} \delta(x_i - x_i^c) dx_i^c \, dx_i \, d\boldsymbol{x}_{-i}$$

$$= \int \prod_{\substack{\ell \in \Delta^{\bar{\mathsf{a}}\mathsf{ug}} \\ \ell \notin \Delta_{ij}^{\mathsf{aug}}}} w_\ell(\boldsymbol{x}_{-i}, x_i)^{[\![ \boldsymbol{x}_{-i}, x_i \models \ell ]\!]} \, .$$

$$\left( \int \prod_{\ell \in \Delta_{ij}^{\mathsf{aug}}} w_\ell(x_i^c, x_j)^{[\![ x_i^c, x_j \models \ell ]\!]} \delta(x_i - x_i^c)[\![ \boldsymbol{x}_{-i}, x_i, x_i^c \models f(\mu) ]\!] dx_i^c \right) dx_i \, d\boldsymbol{x}_{-i}$$

Notice that by the property of Dirac Delta function and the construction of function $f$, it holds that

$$\int \prod_{\ell \in \Delta_{ij}^{\mathsf{aug}}} w_\ell(x_i^c, x_j)^{[\![ x_i^c, x_j \models \ell ]\!]} \delta(x_i - x_i^c)[\![ \boldsymbol{x}_{-i}, x_i, x_i^c \models f(\mu) ]\!] dx_i^c = \prod_{\ell \in \Delta_{ij}} w(x_i, x_j)^{[\![ x_i, x_j \models \ell ]\!]}[\![ \boldsymbol{x} \models \mu ]\!]$$

Therefore, it holds that

$$Z_{f(\mu)}^{\mathsf{aug}} = \int [\![ \boldsymbol{x} \models \mu ]\!] \prod_{\substack{\ell \in \Delta^{\bar{\mathsf{a}}\mathsf{ug}} \\ \ell \notin \Delta_{ij}^{\mathsf{aug}}}} w_\ell(\boldsymbol{x}_{-i}, x_i)^{[\![ \boldsymbol{x}_{-i}, x_i \models \ell ]\!]} \prod_{\ell \in \Delta_{ij}} w_\ell(x_i, x_j)^{[\![ x_i, x_j \models \ell ]\!]} d\boldsymbol{x} = Z_\mu$$

Finally, we have that the WMI of the original model $(\Delta, \mathcal{W})$ equates that of the augmented model $(\Delta^{\mathsf{aug}}, \mathcal{W}^{\mathsf{aug}})$ by observing that $\mathsf{WMI}(\Delta, \mathcal{W}) = \sum_\mu Z_\mu = \sum_{f(\mu)} Z_{f(\mu)}^{\mathsf{aug}} = \mathsf{WMI}(\Delta^{\mathsf{aug}}, \mathcal{W}^{\mathsf{aug}})$.

Moreover, for any univariate literal $\ell$, it can be shown by similar arguments that $\mathsf{WMI}(\Delta \wedge \ell, \mathcal{W}) = \mathsf{WMI}(\Delta^{\mathsf{aug}} \wedge \ell, \mathcal{W}^{\mathsf{aug}})$. Thus, it holds that $\mathsf{Pr}_\Delta(\ell) = \mathsf{WMI}(\Delta \wedge \ell, \mathcal{W})/\mathsf{WMI}(\Delta, \mathcal{W}) = \mathsf{WMI}(\Delta^{\mathsf{aug}} \wedge \ell, \mathcal{W}^{\mathsf{aug}})/\mathsf{WMI}(\Delta^{\mathsf{aug}}, \mathcal{W}^{\mathsf{aug}}) = \mathsf{Pr}_{\Delta^{\mathsf{aug}}}(\ell)$. $\qquad\square$

## B.4 THEOREM 4.3

For the remaining WMI model $(\Delta^{\mathsf{rem}}, \mathcal{W}^{\mathsf{rem}})$, it holds that

$$\begin{aligned}
\mathsf{Pr}_{\Delta^{\mathsf{rem}}}(\ell_{k,i} \wedge \ell_{k,i}^c) &= \frac{\mathsf{WMI}(\Delta^{\mathsf{rem}} \wedge \ell_{k,i} \wedge \ell_{k,i}^c, \mathcal{W}^{\mathsf{rem}})}{\mathsf{WMI}(\Delta^{\mathsf{rem}}, \mathcal{W}^{\mathsf{rem}})} \\
&= \frac{\mathsf{WMI}(\Delta^{\mathsf{rem}} \wedge \ell_{k,i} \wedge \ell_{k,i}^c, \mathcal{W}^{\mathsf{rem}})}{\mathsf{WMI}(\Delta^{\mathsf{rem}} \wedge \ell_{k,i} \wedge \ell_{k,i}^c, \mathcal{W}^{\mathsf{rem}}) + \mathsf{WMI}(\Delta^{\mathsf{rem}} \wedge \neg\ell_{k,i} \wedge \neg\ell_{k,i}^c, \mathcal{W}^{\mathsf{rem}})} \\
&= \frac{\exp{(\theta_{k,i} + \theta_{k,i}^c)}}{r^k + \exp{(\theta_{k,i} + \theta_{k,i}^c)}}
\end{aligned}$$

By substituting the sum of $\theta_{k,i}$ and $\theta_{k,i}^c$ with the first equality in Equation 3, it holds that $\mathsf{Pr}_{\Delta^{\mathsf{rem}}}(\ell_{k,i} \wedge \ell_{k,i}^c) = \mathsf{Pr}_{\Delta^{\mathsf{rel}}}(\ell_{k,i})$; similarly, by substituting the sum with the second equality, it holds that $\mathsf{Pr}_{\Delta^{\mathsf{rem}}}(\ell_{k,i} \wedge \ell_{k,i}^c) = \mathsf{Pr}_{\Delta^{\mathsf{rel}}}(\ell_{k,i}^c)$, which finishes the proof.

# C  Algorithms

---

**Algorithm 2** augmentModel($\Delta, \mathcal{W}, \mathcal{E}_d$)

---

**Input:** a WMI model with SMT formula $\Delta$ and per-literal weights $\mathcal{W}$ and a set $\mathcal{E}_d$ of edges to be deleted

**Output:** augmented WMI model ($\Delta^{\mathsf{aug}}, \mathcal{W}^{\mathsf{aug}}$) and equivalence constraint set $\mathcal{L}$

```
 1: Δ^aug ← copy(Δ)
 2: W^aug ← copy(W)
 3: L ← {}
 4: for edge X_i − X_j ∈ E_d do
 5:     X_i^c ← copy(X_i)                                    ▷ Assume to copy X_i
 6:     ℓ̂ ← (X_i = X_i^c)
 7:     L ← L ∪ {ℓ̂}
 8:     Δ' ← Δ^aug ∧ ℓ̂,
 9:     w_ℓ̂ := δ(X_i, X_i^c)
10:     W^aug ← W^aug ∪ {w_ℓ̂}
11:     for clause Γ ∈ Δ_{i,j} do                            ▷ Rename edges
12:         Γ' ← Γ{X_i : X_i^c}
13:         Δ' ← Δ'{Γ : Γ'}
14:         for each literal ℓ ∈ Γ do
15:             ℓ' ← ℓ{X_i : X_i^c}
16:             w_ℓ' ← copy(w_ℓ)
17:             W^aug ← W^aug ∪ {w_ℓ'} \ {w_ℓ}
18:     for clause Γ ∈ Δ_i do                   ▷ Copy and rename bounding-box literals
19:         Γ' ← copy(Γ)
20:         Δ' ← Δ' ∧ Γ'{X_i : X_i^c}
21:     Δ^aug ← Δ'
22: return Δ^aug, W^aug, L
```

---

---

**Algorithm 3** relaxModel($\Delta^{\text{aug}}, \mathcal{W}^{\text{aug}}, \mathcal{L}$)

---

**Input:** an augmented WMI model ($\Delta^{\text{aug}}, \mathcal{W}^{\text{aug}}$), $\mathcal{L}$: equivalence constraints to be relaxed
**Output:** a relaxed WMI model ($\Delta^{\text{rel}}, \mathcal{W}^{\text{rel}}$), and its "remaining-part" model ($\Delta^{\text{rem}}, \mathcal{W}^{\text{rem}}$).

1: $\Delta^{\text{rem}} \leftarrow \top$
2: $\mathcal{W}^{\text{rem}} \leftarrow \{\}$
3: $\Delta^{\text{rel}} \leftarrow \text{copy}(\Delta^{\text{aug}})$
4: $\mathcal{W}^{\text{rel}} \leftarrow \text{copy}(\mathcal{W}^{\text{aug}})$
5: **for each** $\ell^* : (X_i = X_i^c) \in \mathcal{L}$ **do**
6:     **for** clause $\Gamma \in \Delta_i$ **do**
7:         $\Delta^{\text{rem}} \leftarrow \Delta^{\text{rem}} \wedge \Gamma \wedge \Gamma\{X_i : X_i^c\}$
8:         **for each** literal $\ell \in \Gamma$ **do**
9:             $\ell' \leftarrow \ell\{X_i : X_i^c\}$
10:             $w_{\ell'} \leftarrow \text{copy}(w_\ell)$
11:             $\mathcal{W}^{\text{rel}} \leftarrow \mathcal{W}^{\text{rel}} \cup \{w_{\ell'}\}$
12:             $\mathcal{W}^{\text{rem}} \leftarrow \mathcal{W}^{\text{rem}} \cup \{w_\ell, w_{\ell'}\}$
13:     $\Delta^{\text{rel}} \leftarrow \Delta^{\text{rel}}\{\ell^* : \top\}$                          ▷ disconnect $X_i$ and copy $X_i^c$
14:     $\mathcal{W}^{\text{rel}} \leftarrow \mathcal{W}^{\text{rel}} \setminus \{w_{\ell^*}\}$
15:     $\Delta^{\text{rem}} \leftarrow \Delta^{\text{rem}} \wedge \ell^*$
16:     $\mathcal{W}^{\text{rem}} \leftarrow \mathcal{W}^{\text{rem}} \cup \{w_{\ell^*}\}$
17: **return** ($\Delta^{\text{rel}}, \mathcal{W}^{\text{rel}}$), ($\Delta^{\text{rem}}, \mathcal{W}^{\text{rem}}$)

---

---

**Algorithm 4** addingCompensations($\Delta^{\text{rel}}, \mathcal{W}^{\text{rel}}, \mathcal{L}, K$)

---

**Input:** a relaxed WMI model ($\Delta^{\text{rel}}, \mathcal{W}^{\text{rel}}$), $K$ number of compensating literals to introduce
**Output:** the relaxed WMI model ($\Delta_+^{\text{rel}}, \mathcal{W}_+^{\text{rel}}$) with compensating literals initialized.

1: $\Delta_+^{\text{rel}} \leftarrow \Delta^{\text{rel}}, \mathcal{W}_+^{\text{rel}} \leftarrow \mathcal{W}^{\text{rel}}$
2: $\mathbf{X}_o \leftarrow \text{nonCopyVars}(\mathcal{L})$                         ▷ Gather original variables
3: **for each** $X_i \in \mathbf{X}_o$ **do**
4:     **for** $k = 1, \ldots, K$ **do**
5:         $\tau_{i,k} \sim \text{Uniform}(\text{support}(X_i))$            ▷ Randomly help support
6:         $\sigma_{i,k} \sim \text{Uniform}(\{+1, -1\})$               ▷ And pick one half
7:         $\ell_{i,k} \leftarrow (X_i \le \sigma_{i,k} \cdot \tau_{i,k})$
8:         $\Delta_{+,i}^{\text{rel}} \leftarrow \Delta_{+,i}^{\text{rel}} \wedge \ell_{i,k}$
9:         $\theta_{i,k} \leftarrow 1$                                    ▷ Initiate potentials
10:         $w_{\ell_{i,k}} := \exp(\theta_{i,k})$
11:         $\mathcal{W}_+^{\text{rel}} \leftarrow \mathcal{W}_+^{\text{rel}} \cup \{w_{\ell_{i,k}}\}$
12:         **for each** $\ell : (X_i = X_i^c) \in \mathcal{L}$ **do**
13:             $\ell_{i,k}^c \leftarrow (X_i^c \le \sigma_{i,k} \cdot \tau_{i,k})$
14:             $\Delta_{+,i}^{\text{rel},c} \leftarrow \Delta_{+,i}^{\text{rel},c} \wedge \ell_{i,k}$
15:             $\theta_{i,k}^c \leftarrow 1$                           ▷ Initiate potentials
16:             $w_{\ell_{i,k}^c} := \exp(\theta_{i,k}^c)$
17:             $\mathcal{W}_+^{\text{rel}} \leftarrow \mathcal{W}_+^{\text{rel}} \cup \{w_{\ell_{i,k}^c}\}$
18: **Return** ($\Delta_+^{\text{rel}}, \mathcal{W}_+^{\text{rel}}$)

---