[Reviews · NeurIPS 2020]

Review 1

Summary and Contributions: - Complexity results on the weighted model integration - A new approach based on relaxing-compensating-recoverying idea from message passing in graphical models - Experiments showing the proposed approach achieves state-of-the-art performance

Strengths: - Solid work - Combination of theoretical and practical analysis of the problem - Sophisticated approach with SOTA performance - Good literature review

Weaknesses: - Motivation lacks a more direct connection with the problem at hand (how do we go from self-driving cars to integrating over intervals?) - Presentation require knowledgeable reader and relies heavily on supp. material - Experiments with artificial toy problems

Correctness: The theoretical results appear to be sound; the empirical methodology is correct.

Clarity: The paper is very dense, but the text is well-written. Some examples here and there could help introduce and engage the less knowledgeable reader.

Relation to Prior Work: The contributions are clearly presented, and related work is properly reviewed.

Reproducibility: Yes

Additional Feedback: In line 98, I believe Tb should take values in [0,1], not [-1,1].


Review 2

Summary and Contributions: This paper focuses on the problem of weighted model integration. The paper has two components: theoretical hardness results and an abstraction based approach. The theoretical hardness claims that inference when tree width is 1 but diameter is n or when treewidth is 2 but diameter is log n is hard. The empirical part of the paper focuses on the rely, compensate and integrate approach which is similar to lot of work recently in counting community (see references below). The empirical results show promise.

Strengths: WMI is an important problem and the paper does attempt to make contributions along both angles. The empirical results clearly demonstrate progress over state of the art.

Weaknesses: There are two weaknesses with the paper: 1. I am not sure if the theoretical claims are correct. They seem very strong as only in propositional case, we have FPT for constant treewidth, so the claims need to talk about what makes problem so hard when you add in continuous variable. I looked at the proof and there are several things that trouble me: There is change of representation; subset sum is #P-complete when we are concerned with binary representation otherwise we have pseudo-polynomial time algorithms by dynamic programming. The proofs seem to work on integer representation not binary representation as the treewidth for binary representation is still n. The second issue is lack of relevant context to series of works on usage of abstraction to tackle the high treewidth case in the context of propositional formulas. I have listed references below. I think it is still okay for authors to claim that the prior work focuses on propositional case but their contribution focuses on the WMI. [1] Dell, H., Roth, M., Wellnitz, P.: Counting answers to existential questions. In: ICALP 2019. LIPIcs, vol. 132, pp. 113:1–113:15. Schloss Dagstuhl - Leibniz-Zentrum für Informatik (2019) [2] Eiben, E., Ganian, R., Hamm, T., Kwon, O.: Measuring what matters: a hybrid approach to dynamic programming with treewidth. In: MFCS 2019. LIPIcs, vol. 138, pp. 42:1–42:15. Dagstuhl Publishing (2019) [3] Ganian, R., Ramanujan, M.S., Szeider, S.: Combining treewidth and backdoors for CSP. In: STACS 2017, pp. 36:1–36:17 (2017). https://doi.org/10.4230/LIPIcs.STACS.2017.36 [4] Hecher, M., Morak, M., Woltran, S.: Structural decompositions of epistemic logic programs. CoRR abs/2001.04219 (2020). http://arxiv.org/abs/2001.04219

Correctness: I am unsure about the correctness of theoretical results.

Clarity: The paper is well written on high level but since it makes a major claim, it should give intuition about the correctness of the proof.

Relation to Prior Work: Relevant references are missing as I have pointed above. It would be good if authors can add them so as to provide proper context to the reader.

Reproducibility: No

Additional Feedback: Thank you for the response. During the discussion phase, other reviewers clarified the reduction which makes me believe that the claim is probably correct. Since I have not been able to check details precisely, I think it is right from scientific perspective (given that these reviews would be public) for me to reserve full endorsement. I will strongly encourage authors to provide a rigorous proof of the claim; in the sense, clearly specifying the reduction from subset sum in binary representation to WMI in PTIME (taking care of the representation of the constraints). I have increased the score from 3 to 5 to reflect my opinion after the PC discussion.


Review 3

Summary and Contributions: This paper presents theoretical and algorithmic results about the WMI framework. While the theoretical results are negative in the sense that they show the limits of the tractability of the problems solved by the WMI framework, the authors then present an algorithmic approach to overcome this limitation: an approximate WMI solver.

Strengths: Both theoretical and algorithmic results are very important for the concepts around WMI. The experimentation section if very clear and focus both on the questions answered by these experiences and the methods used.

Weaknesses: A minor issue that the Watts-Strogatz graphs are used in the experimentation section without any justification.

Correctness: To the best of my knowledge and understanding, I think that the claims of this paper are correct.

Clarity: Yes the paper is very well written. One typo in the conclusion: ourl -> our.

Relation to Prior Work: The relation to prior work is well defined.

Reproducibility: Yes

Additional Feedback:

[Author Response · NeurIPS 2020]

We thank reviewers for their feedback and for agreeing on the importance of our work under both a theoretical and practical perspective [all reviewers (Rs)], for appreciating the way it is written and posed in the context of the hybrid probabilistic inference literature [R2 and R4] and for showing a clear improvement over SOTA WMI solvers [all Rs]. In the following, we will address the individual concerns raised.

**\*R2\***

**[Presentation& Motivation]** We will provide in the camera-ready an additional example of WMI, illustrate in detail an execution of ReCoIn and better substantiate how integration in hybrid domains is at the core of probabilistic reasoning in real applications as well (e.g., in visual scene grounding).

**[Integration in [-1, 1]]** The interval of integration in the paper is correct and follows from the reduction to a WMI problem with only continuous variables, as illustrated in Appendix A.

**\*R3\***

**[Hardness with continuous variables]** Indeed, the introduction of continuous variables increases the hardness of probabilistic inference. To see *why* this happens in the context of message passing, it has been noted in [1] that representing messages over SMT(LRA) theories induces a piecewise representation of the involved densities. From here one can see how inference complexity depends on the number of these pieces – which is exponential in the diameter of a primal graph and is not being bounded by its treewidth. This is strikingly different from discrete domains, where messages can be represented as finite tables whose sizes depend only on the treewidth of the model given a fixed variable scope. We point out that this increase in complexity is not peculiar to WMI inference alone and has been already observed in the probabilistic graphical model community. As a simple case, consider conditional linear Gaussian (CLG) models [2]. It is known that even if the graph structure of a CLG is restricted to have treewidth 1 integration for marginal inference can become NP-hard and even approximate inference routines with guarantees on these simple networks can be intractable.

**[Binary representation of #SSP]** In our hardness proof, we adopted the binary representation for the integers – as commonly assumed in reductions of this kind [2,3] – which makes the decision version of problem NP-complete and the Subset Sum Problem (#SSP) #P-complete. We will make this assumption explicit in the text. If instead the (non-standard) unary representation were used for integers for the #SSP, then the problem would be in P as the reviewer says. It could be interesting future work to prove hardness of WMI with rational unary numbers, or PTime complexity of integer unary parameters, neither of which where the goal of our paper. Note that adopting the usual binary representation for integers ***does not change the treewidth*** of the primal graph of the formula used in our reduction. This is due to the fact that the edges in the primal graph are defined by the dependencies in the formula which in turn is not affected by a change of representation of the constants in the SMT literals.

**[Missing references]** We thank the reviewer for pointing us to some recent advances in counting in discrete domains and in the propositional case. We will properly contrast our work to them in the camera-ready. Here we point out that they are not directly applicable to our WMI setting and that out setting is inherently more challenging than the settings mentioned in the linked references. In fact, we assume an SMT(LRA) theory involving continuous and discrete variables and a first order representation to be given. We do not build any further abstraction as in this context it would require going to a higher-order logical representation or exploit some symmetries in the theory, as usually done in lifted probabilistic inference. Lastly, to adopt one of the techniques referenced would require to revert to a fully discrete representation. That is, discretizing a WMI problem in both the support and densities and essentially reducing it to a weighted model counting problem. A discretization of this kind would be quite problematic, however: binning continuous supports is highly task-dependent and the loss in "resolution" can greatly hurt the quality of inference and potentially increase exponentially the problem representation.

**\*R4\***

**[Watts-Strogatz graphs]** We employ Watts-Strogatz (WS) graphs as benchmarks with an increasing number of variables with bounded treewidth larger than 1. In the final paper, we will highlight these properties of WS graphs and add experiments on other challenging topologies (e.g., cactus graphs) where we can observe the same solver behavior.

**[Typo]** Thanks for spotting the typo, we will fix it in the camera-ready and proof read the paper again.

[1] Zeng et al. "Scaling up Hybrid Probabilistic Inference with Logical and Arithmetic Constraints via Message Passing", ICML 2020.

[2] Lerner, et al. "Inference in hybrid networks: theoretical limits and practical algorithms." UAI 2001.

[3] De Campos, C. P. "New complexity results for MAP in Bayesian networks." IJCAI. 2011.


[Meta-Review · NeurIPS 2020]

Two reviewers rated the paper very highly (8, 9). R3 initially gave a very low rating and expressed concerns about correctness of the intractability result, especially related to the representation of numbers in the reduction from subset sum. This point received significant discussion. The meta-reviewer read the proof and felt it was very clear and agrees with the authors. There is one minor ambiguity that can be resolved: the authors don’t specify the representation of the constants that appear in the constraints in the WMI problem instance. It was assumed these should be rational numbers; in this case the constants constructed during the reduction for the constraints of the WMI problem have bit complexity proportional to that of the numbers that appear in the subset sum instance. Given that the only major critique was resolved during discussion, the meta-reviewer recommends accept. R3 expressed some hesitation, but raised their score from 3 to 5.